# Tumour gene expression signature in primary melanoma predicts long-term outcomes

Manik Garg [1], Dominique-Laurent Couturier [2], Jérémie Nsengimana [3,4], Nuno A. Fonseca [5], Matthew Wongchenko [6], Yibing Yan[6], Martin Lauss[7], Göran B. Jönsson[7], Julia Newton-Bishop[3], Christine Parkinson[8], Mark R. Middleton [9], D. Timothy Bishop [3], Sarah McDonald[10], Nikki Stefanos[10], John Tadross [10], Ismael A. Vergara [11,12], Serigne Lo [11,12,13], Felicity Newell [14], James S. Wilmott[11,12], John F. Thompson [11,12,15], Georgina V. Long[11,12,16], Richard A. Scolyer [11,12,17], Pippa Corrie[8], David J. Adams [18,19], Alvis Brazma [1,19] & Roy Rabbie [8,18,19✉]

Adjuvant systemic therapies are now routinely used following resection of stage III melanoma, however accurate prognostic information is needed to better stratify patients. We use differential expression analyses of primary tumours from 204 RNA-sequenced melanomas within a large adjuvant trial, identifying a 121 metastasis-associated gene signature. This signature strongly associated with progression-free (HR = 1.63, $p = 5.24 \times 10^{-5}$) and overall survival (HR = 1.61, $p = 1.67 \times 10^{-4}$), was validated in 175 regional lymph nodes metastasis as well as two externally ascertained datasets. The machine learning classification models trained using the signature genes performed significantly better in predicting metastases than models trained with clinical covariates ($p_{AUROC} = 7.03 \times 10^{-4}$), or published prognostic signatures ($p_{AUROC} < 0.05$). The signature score negatively correlated with measures of immune cell infiltration ($\rho = -0.75$, $p < 2.2 \times 10^{-16}$), with a higher score representing reduced lymphocyte infiltration and a higher 5-year risk of death in stage II melanoma. Our expression signature identifies melanoma patients at higher risk of metastases and warrants further evaluation in adjuvant clinical trials.

[1] European Molecular Biology Laboratory, European Bioinformatics Institute (EMBL-EBI), Hinxton, Cambridgeshire, UK. [2] Cancer Research UK Cambridge Institute, University of Cambridge, Li Ka Shing Centre, Robinson Way, Cambridge, UK. [3] University of Leeds School of Medicine, Leeds, United Kingdom. [4] Biostatistics Research Group, Population Health Sciences Institute, Faculty of Medical Sciences, Newcastle University, Newcastle upon Tyne, UK. [5] CIBIO/InBIO-Centro de Investigação em Biodiversidade e Recursos Genéticos, Universidade do Porto, Rua Padre Armando Quintas, 4485-601 Vairão, Portugal. [6] Oncology Biomarker Development, Genentech Inc., 1 DNA Way, South San Francisco, CA 94080, USA. [7] Lund University Cancer Center, Lund University, Lund, Sweden. [8] Cambridge Cancer Centre, Cambridge University Hospitals NHS Foundation Trust, Cambridge, UK. [9] Oxford NIHR Biomedical Research Centre and Department of Oncology, University of Oxford, Oxford, UK. [10] Department of Pathology, Cambridge University Hospitals NHS Foundation Trust, Cambridge, UK. [11] Melanoma Institute Australia, The University of Sydney, North Sydney, NSW, Australia. [12] Faculty of Medicine and Health, The University of Sydney, Sydney, NSW, Australia. [13] Institute for Research and Medical Consultations (IRMC), Imam Abdulrahman Bin Faisal University, Dammam, Saudi Arabia. [14] QIMR Berghofer Medical Research Institute, Brisbane, QLD, Australia. [15] Discipline of Surgery, Faculty of Medicine and Health, The University of Sydney, Sydney, NSW, Australia. [16] Royal North Shore and Mater Hospitals, Sydney, Australia. [17] Tissue Pathology and Diagnostic Oncology, Royal Prince Alfred Hospital and New South Wales Health Pathology, Sydney, NSW, Australia. [18] Experimental Cancer Genetics, The Wellcome Sanger Institute, Hinxton, Cambridgeshire, UK. [19]These authors contributed equally: David J. Adams, Alvis Brazma, Roy Rabbie. ✉email: rr13@sanger.ac.uk

Cutaneous melanoma (CM) accounts for 75% of skin cancer-related deaths, and the incidence has been increasing worldwide[1]. Most patients present with primary tumours and the majority will be cured by local surgery. Outcomes for patients with metastatic melanoma have improved radically over the past 10 years with the introduction of new systemic therapies[2], although median survival of patients has remained at ~3 years. Importantly, of those patients who ultimately die of melanoma a significant proportion originally presented with early-stage disease[3], suggesting that there is a subgroup of these patients who have aggressive tumours. Thus, optimal management of early melanoma is key to improving outcomes.

Patients with resected AJCC stage III melanoma are now eligible for adjuvant immune checkpoint inhibitors, as well as *BRAF*-targeted therapies, based on randomised trials confirming a reduction in the risk of relapse and improved overall survival (OS)[4–7]. Clinical trials are underway to evaluate similar therapies in resected stage IIB/C patients[8], whose outcomes reflect that of untreated stage IIIA/B melanoma[9]. As such, the number of patients eligible for the treatment with adjuvant therapies over the coming years is expected to increase substantially. These modern anti-cancer drugs are high cost and carry a risk of both life-changing and life-threatening toxicities, so there is a growing desire to more accurately predict those patients at high risk of recurrence in whom intervention is expected to be beneficial so-as-to avoid over-treating patients likely to have been cured of their disease by surgery alone.

Gene expression signatures have the potential to improve the prediction of the biological behaviour of melanoma by objectively defining "high risk" on a molecular level[10]. Previous transcriptomic analyses of CM identified patterns of gene expression associated with survival independent of AJCC stage[11]. Building on these data, Gerami et al. first reported a proprietary prognostic gene expression profile (GEP) test utilising a 27-gene panel for use in patients with CM (Gerami_27)[12]. The test uses quantitative reverse transcriptase polymerase chain reaction technology to measure the expression of individual genes from formalin-fixed paraffin-embedded (FFPE) primary melanomas to provide a binary classification of low (class 1) or high (class 2) risk for developing metastases within 5 years of diagnosis (with A and B subclasses to further stratify risk)[13]. The signature's performance has since been evaluated in a number of retrospective clinical studies evaluating recurrence-free survival[14,15]. A further recent unsupervised clustering analysis based on 677 primary melanoma transcriptomes (generated using the Illumina DASL array platform) embedded within a population-controlled cohort study from the Leeds Melanoma Cohort (LMC) identified a six-class 150 gene prognostic signature (herein referred to as LMC_150)[16]. The signature uniquely demonstrated prognostic relevance (melanoma-specific survival; MSS) in patients with stage I primary melanoma and further predicted poor outcomes in patients undergoing immunotherapy[16]. Heterogeneity in key aspects of the aforementioned studies (including varying trial design, sample type, sequencing platforms and primary outcome measures) may partially explain the small number of overlapping genes (n = 4) between both sets of signatures. Furthermore, owing to a lack of prospective data proving the clinical utility of such prognostic molecular tools[17], there are currently no established prognostic biomarkers able to accurately identify truly high-risk patients.

Using patient samples and long-term clinical outcome data from one of the largest adjuvant melanoma trials[18,19], we undertook RNA sequencing of the primary tumour matched with robust prospective clinical data to uncover a molecular signature that could be used to predict patient outcomes. This was then validated in two externally ascertained datasets.

## Results

**Prognostic signature generated using covariate-corrected differential expression**. The structure of the datasets and analyses are depicted in Supplementary Fig. S1. Principal component analysis (PCA) showed that primary CMs (n = 204) and melanoma spread to local lymph nodes (LNs; n = 175) clustered separately, suggesting an impact of the microenvironment on tumour gene expression (Supplementary Fig. S2; see "Methods" section "Visualization of inherent distribution of samples"). We therefore decided to treat these as separate datasets, focussing our analyses on the primary melanoma samples followed by a validation of our results in the regional LN metastases from this dataset. We conducted a differential expression analysis, identifying differences in gene expression levels in primary tumours between those patients with and without distant metastasis over a minimum of 6 years follow up, while controlling for a key set of variables that were independently associated with distant metastases, including Stage (AJCC 7th edition[20], herein referred to as "stage"), Breslow thickness, ECOG; Eastern Cooperative Oncology Group Performance Status and the experimental adjuvant therapy (Supplementary Fig. S3a and Supplementary Table S1; see "Methods" sections "Clinical covariate selection" and "Differential expression analysis"). Our analyses revealed 197 significantly differentially expressed genes (DEGs, FDR-adjusted p value <0.1) associated with metastases (Supplementary Figs. S1 and S3b). These DEGs were further filtered to remove pseudogenes (n = 39) and those genes not identified within the LMC DASL array (n = 37)[16] to enable external validation of our signature (Supplementary Fig. S1). We were therefore left with 121 DEGs, which made up our core prognostic signature herein referred to as "Cam_121" (Supplementary Data 1).

**Signature added incremental prognostic value when combined with conventional clinical staging**. In order to explore the relationship between Cam_121 gene expression and prognosis, we first performed univariate Cox regression using the weighted Cam_121 expression score (see "Methods" section "Survival analyses") as a predictor. We found that the Cam_121 signature significantly associated with both OS (hazard ratio (HR) = 1.64 (95% CI 1.30–2.07), $p = 3.56 \times 10^{-5}$) and progression-free survival (PFS; HR = 1.63 (95% CI 1.31–2.02), $p = 8.92 \times 10^{-6}$; Fig. 1). In order to evaluate whether the signature score contributed independent prognostic information while controlling for conventional clinical covariates, multivariate Cox regression analyses were performed (Fig. 1c). The signature score was significantly associated with both OS (HR = 1.61 (95% CI 1.26–2.07), $p = 0.000167$) and PFS (HR = 1.63 (95% CI 1.29–2.07), $p = 5.24 \times 10^{-5}$) in multivariate Cox regression models.

In order to avoid the risk of overfitting, we further tested the performance of Cam_121 in an (entirely separate) set of samples from regional LN metastasis embedded within this dataset (n = 143). We found that the weighted signature score was also significantly associated with both OS (HR = 1.72 (95% CI 1.37–2.14), $p = 1.53 \times 10^{-6}$) and PFS (HR = 1.75 (95% CI 1.43–2.16), $p = 1.10 \times 10^{-7}$) in multivariate Cox regression models (Fig. 2a–c). Thereby indicating that Cam_121 could also be relevant as a prognostic tool after the resection of stage III (regional LN positive) melanoma.

We further tested the signatures' prognostic power in four externally acquired independent validation datasets from; (i) the LMC (n = 677)[16]; (ii) The Cancer Genome Atlas (TCGA-SKCM[21]; skin = 159 and LN = 216); (iii) the Lund Primary Melanoma Cohort[22] (n = 223) and (iv) the Australian Melanoma Genome Project[23]. Validation within the LMC confirmed that

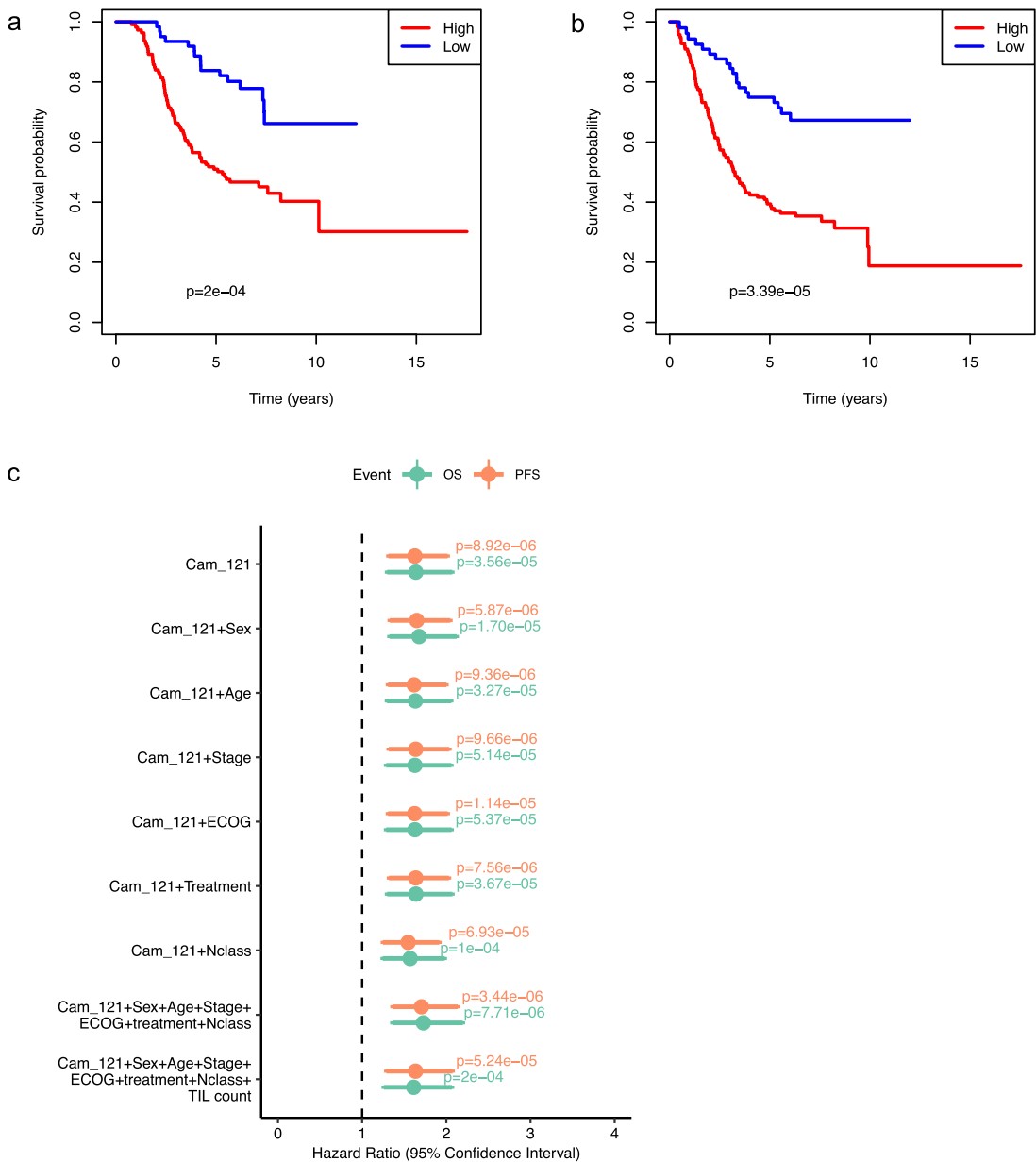

**Fig. 1 The Cam_121 gene expression signature is strongly associated with survival in uni- and multivariate Cox regression analyses (AVAST-M primary melanoma cohort; $n = 194$).** Kaplan–Meier survival plots comparing the survival probabilities (*y*-axes) as a function of time in years (*x*-axes) of groups with high and low "Cam_121" (quantile 0.33 split) for outcomes **a** overall survival (OS) and **b** progression-free survival (PFS). The *p* value of a two-sided logrank test comparing the survival distributions of both groups are indicated. **c** Forest plot indicating the hazard ratio (HR) estimates relating to the Cam_121 signature when predicting OS (green) and PFS (orange) by means of Cox proportional hazard models controlling for different (sets of) clinical variables (*y*-axis). The HR estimates are indicated by the dots at the centre of the error bars; the horizontal error bars correspond to the 95% confidence intervals of the HR. The two-sided Wald *t* test *p* values corresponding to the signature "Cam_121" parameter are indicated for each model and outcome. ECOG Eastern Cooperative Oncology Group Performance Status. TIL count Tumour-infiltrating lymphocyte count.

Cam_121 was associated with melanoma-specific survival in both univariate (HR = 1.49 (95% CI 1.27–1.74), $p = 5 \times 10^{-7}$) and multivariate Cox regression models (HR = 1.7 (95% CI), $p = 0.001$, Fig. 2d, e). Owing to a lack of power when considering true primary melanomas within TCGA-SKCM dataset ($n = 87$), samples from primary tumours were considered together with regional cutaneous relapsed tumours (defined herein as "Skin TCGA-SKCM" ($n = 159$)) and were tested separately from the regional LN samples ($n = 216$). Cam_121 was associated with OS in both univariate (skin: HR = 1.50, 95% CI = (1.15, 1.96), $p = 0.00273$; LN: HR = 1.28, 95% CI = (1.06, 1.56), $p = 0.0109$) and

multivariate survival analyses (skin: HR = 1.59, 95% CI = (1.17, 2.17), $p = 0.00348$; LN: HR = 1.28, 95% CI = (1.03, 1.59), $p = 0.0256$) in these two external datasets (Fig. 2f–i). A third external validation was attempted using the Lund Primary Melanoma Cohort[22] ($n = 223$); however (owing to the use of less sensitive sequencing technologies within the study), only 24 of the Cam_121 genes were identified within this dataset (Supplementary Fig. S5b). Nonetheless, this 24-gene signature was significantly correlated with PFS (HR = 1.67 (95% CI 1.06–2.62), $p = 0.03$ univariate Cox regression analyses), but not with OS ($p = 0.32$) (Supplementary Fig. S5a). A final validation was attempted

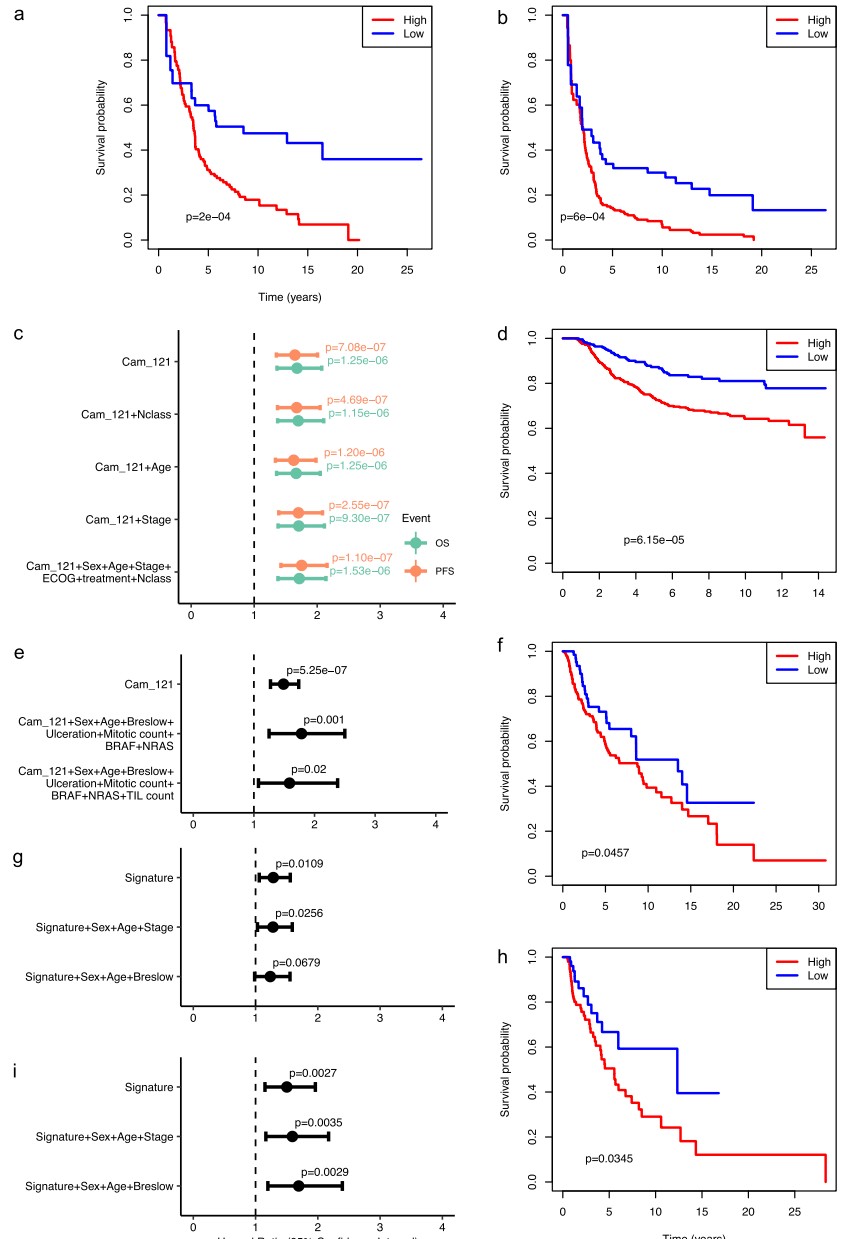

**Fig. 2 Validation of the Cam_121 in further datasets.** We explored the Cam_121 signature in the entirely separate lymph node (LN) samples from the AVAST-M Cohort (**a–c**; $n = 143$), as well as in three independently acquired external datasets including the Leeds Melanoma Cohort (**d**, **e**; $n = 677$), TCGA-SKCM LN (**f**, **g**; $n = 212$) and TCGA-SKCM Skin (**h**, **i**; $n = 156$). AVAST-M LN Cohort: Kaplan–Meier (K–M) survival plots comparing the survival probabilities (y-axes) as a function of time in years (x-axes) of groups with high and low "Cam_121" (quantile 0.33 split) for outcomes **a** overall survival (OS) and **b** progression-free survival (PFS). The *p* value of a logrank test comparing the survival distributions of both groups is indicated on each curve. **c** Forest plot indicating the hazard ratio (HR) estimates (dots at the centre of the error bars) and corresponding 95% confidence intervals (horizontal error bars) related to the Cam_121 signature when predicting OS (green) and PFS (orange) by means of Cox proportional hazard models when controlling for different (sets of) clinical variables (y-axis). The two-sided Wald *t* test *p* values corresponding to the signature "Cam_121" parameter are indicated for each model and outcome. ECOG Eastern Cooperative Oncology Group Performance Status. Leeds Melanoma Cohort: **d** K–M curve comparing the melanoma-specific survival probabilities (y-axis) of groups with high and low "Cam_121" (quantile 0.33 split) through time in years (x-axis) and *p* value of a two-sided logrank test comparing the survival distributions. **e** Forest plot showing the HR estimates (dots at the centre of the error bars) and 95% confidence intervals (horizontal error bars) of the HR estimates corresponding to the continuous signature "Cam_121" parameter when predicting melanoma-specific survival by means of different Cox proportional hazard models (y-axis). Multivariate correction was undertaken for sex, age, Breslow thickness, ulceration, mitotic count, as well as *BRAF* and *NRAS* mutation status and (in the final row), correction was also undertaken for the tumour-infiltrating lymphocyte (TIL) score. TCGA-SKCM Cohort: **f**, **h** K–M curves comparing the overall survival probabilities of groups with high and low "Cam_121" (quantile 0.33 split) and *p* value of a two-sided logrank test comparing the survival distributions in TCGA-SKCM LN and TCGA-SKCM Skin datasets, respectively. TCGA-SKCM Cohort: **g**, **i** Forest plot showing the HR estimates (dots at the centre of the error bars) and 95% confidence intervals (horizontal error bars) corresponding to the of related to the continuous signature "Cam_121" parameter when controlling for different (sets of) clinical variables (y-axes) in the TCGA-SKCM LN and TCGA-SKCM Skin datasets, respectively. Multivariate correction was undertaken for sex, age, stage and Breslow thickness. The *p* values of two-sided Wald *t* tests corresponding to the signature "Cam_121" parameter are indicated for each model and outcome (no multiplicity correction used).

using the Australia Melanoma Genome Project dataset[23], in which only 55 samples from a mixture of tissue sites (including primary tumours, regional LNs, distant metastases, in-transit metastases and others) were available for analysis. Cox regression parameter estimates showed the same trend as observed above when comparing the "high"/"low" risk Cam_121 cohorts in this dataset (Supplementary Fig. S4a), however, significance was not achieved. Our power calculations revealed that the sample size was too small to statistically detect the effect of interest with a high probability (Supplementary Fig. S4b; see power analysis in "Methods" section "Power calculation for the external validation datasets").

The published signature from Gerami et al.[12] (Gerami_27; $n = 27$ genes) was not associated with OS in multivariate models in the AVAST-M primary melanoma dataset. The signature from Thakur et al.[16] (LMC_150; $n = 150$ genes) was associated with both OS and PFS, though the wide confidence intervals may in part be reflective of a higher proportion of stage III patients in the AVAST-M dataset (Supplementary Fig. S6).

**Cam_121 predicts metastasis better than both clinical covariates and published prognostic signatures**. We further sought to determine whether the Cam_121 gene expression signature outperforms key clinical covariates in predicting whether a primary melanoma would ultimately metastasise to distant body sites or not. For this, we developed separate machine learning (ML) classification models using the Cam_121 gene expression values as features, as well as using clinical covariates as features, with the aim of maximising the area under the sensitivity vs (1-specificity) curve (herein referred to as "AUROC" (area under the receiver operating characteristic curve)) as both the metrics are important in this case (see "Methods" section "Machine learning analysis"; Supplementary Fig. S15). Note that these clinical covariates were selected independently based on their level of association with distant metastases (Supplementary Table S1a). We found that models trained with the Cam_121 gene expression signature as features significantly outperformed the models trained with the clinical covariates as features ($p_{AUROC} = 2.27 \times 10^{-3}$, $p_{sensitivity} = 1.79 \times 10^{-3}$, $p_{specificity} = 0.46$; Supplementary Table S2a), and this remained consistent across all ML classifiers (Fig. 3a, Supplementary Figs. S7 and S8, and Supplementary Table S2a). The classifier giving the highest AUROC with the prognostic signature gave better results across all three performance metrics than the classifier giving the highest AUROC with the clinical covariates alone: AUROC ($0.67 \pm 0.12$ with the clinical covariates alone vs $0.83 \pm 0.09$ with the Cam_121 alone), sensitivity ($0.58 \pm 0.16$ vs $0.75 \pm 0.13$) and specificity ($0.71 \pm 0.16$ vs $0.73 \pm 0.14$; Fig. 3a and Supplementary Fig. S8).

In order to reduce the risk of bias that might result from feature selection (of both Cam_121 and clinical covariates) and training/testing from the same dataset, we went on to validate our findings in an entirely independent dataset represented by the regional LN samples from within the AVAST-M dataset ($n = 143$). In this model, the classifiers giving the highest AUROC for each set of features on the training data (AVAST-M primary melanoma data) were selected for further validation within regional LN samples. We found that the classification model developed using the Cam_121 gene expression signature as features (AUROC = 0.67) significantly outperformed the classification model developed using the clinical covariates alone (AUROC = 0.54; DeLong's test $p$ value = 0.02, $z = 2.05$; Fig. 3b and Supplementary Table S2b). In particular, adding the signature to the clinical covariates (Cam_121 + clinical covariates) correctly predicted an additional three overlapping cases that were missed out by the model trained on the signature alone (49 overlapping cases in Fig. 3c vs 46 overlapping cases in Fig. 3d).

In order to test the performance of the published prognostic signatures from Gerami et al.[12] (Gerami_27; $n = 27$ genes) and Thakur et al.[16] (LMC_150; $n = 150$ genes) in predicting metastases in an unbiased way, we repeated the training and testing of classification models in the entirely independent AVAST-M LN dataset ($n = 143$) from which no feature selection has been undertaken. We found that Cam_121 significantly outperformed the baseline clinical covariates (Cam_121 vs clinical covariates: $p_{AUROC} = 7.03 \times 10^{-4}$, $z_{AUROC} = 4.44$), as well as these two published signatures at the 5% significance level (Cam_121 vs LMC_150: $p_{AUROC} = 0.02$, $z_{AUROC} = 2.30$; Cam_121 vs Gerami_27: $p_{AUROC} = 0.012$, $z_{AUROC} = 2.45$; Fig. 3e and Supplementary Table S2c, d).

**Cam_121 gene signature score performed significantly better than genes selected at random in predicting overall and progression-free survival**. In light of reports suggesting that randomly selected genes may perform equally well in predicting prognosis as published signatures[24], we further tested the performance of Cam_121 against a signature of 121 randomly selected genes (from a pool of 19,434 protein-coding genes in our dataset and repeated 1000 times; see "Methods" section "Testing signature performance against randomly selected genes"). Cam_121 significantly outperformed random signatures across all measures of clinical efficacy including: OS ($p = 0.001$) and PFS ($p = 0.001$) in multivariate Cox regression models (Supplementary Fig. S9).

**Stage II patients with a "high-risk" signature demonstrated a 33% risk of death at 5 years, a threshold for which adjuvant therapy could be considered**. We envisage that one of the central clinical applications of a prognostic GEP might be to identify those patients with stage II melanoma who may be at higher risk of relapse or death and for whom adjuvant systemic therapies may be considered. In order to compare our data with the registration adjuvant melanoma trials[4–7], we measured the absolute risk of death at 5 years (calculated as the proportion of patients who died due to melanoma within 5 years from diagnosis; see "Methods" section "Determination of the weighted expression score cut-off to define "high" and "low" absolute risk of death at 5 years").

Analyses within the LMC cohort (where there was a higher preponderance of early-stage patients) revealed that stage II patients had a 27% (76/279) baseline absolute risk of death at 5 years. This risk rose to 33% (64/192) in those stage II patients with a high-risk-weighted Cam_121 expression score profile and dropped to 14% (12/87) in those stage II patients with a low-risk profile (Table 1). The stratification of high/low-risk cohorts in this context was based on a 0.33 quantile cut off of the weighted Cam_121 expression score and subsequent references to high/low Cam_121 risk groups refer to these weighted expression groups.

**Per-gene analyses**. In order to determine the relative influence of each gene and each baseline clinical covariate within the ML model, we analysed their feature importance scores and found that no single feature dominated the performance of the model (Supplementary Fig. S10), suggesting that it is the combination of all the features that yielded improved performance over the baseline clinical covariates (Fig. 3a, and Supplementary Figs. S7 and S8). In keeping with this, none of the Cam_121 genes proved significant in the per-gene multivariate survival analyses after correcting for multiple testing ($p$ value <0.05), further confirming that it is the combination of all the Cam_121 signature genes that provide the improved performance in predicting OS and PFS (Supplementary Figs. S11 and S12).

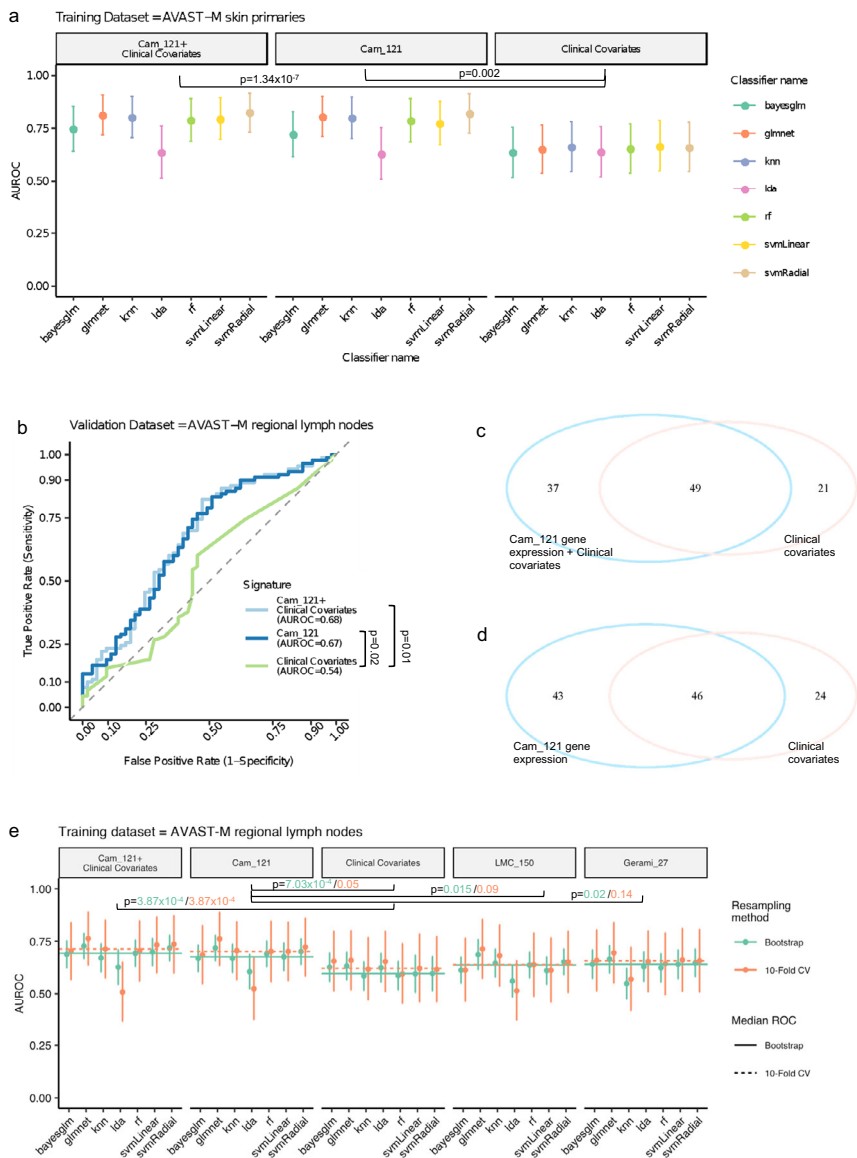

**Fig. 3 The Cam_121 gene expression signature is predictive of metastases across multiple machine learning classification models and in an internal validation dataset (*n* = 143). a** Plot showing the mean ± standard deviation of area under the ROC curve (AUROC) predicted by different classification models when trained using tenfold cross validation (CV; repeats = 1000) on the AVAST-M primary melanoma dataset (*n* = 194). The features used for training each classification model are indicated on the top grey panel. Within each panel, seven different machine learning classifiers were trained to predict metastases. Statistical comparison using one-sided two-sample Welch *t* tests are indicated (see also Supplementary Table S2a). **b** Area under the ROC curve plots, showing, for the best performing classification model selected in each panel of **a**, its performance on an entirely separate AVAST-M lymph node validation dataset (*n* = 143). The one-sided DeLong's test *p* value is reported for each comparison (see also Supplementary Table S2b). **c**, **d** Venn diagrams comparing the number of correctly predicted relapse outcomes (yes/no) of 143 patients from the models described in **b**. **c** Venn diagram showing the number of correctly predicted relapse outcomes specific to or common between "Cam_121 + clinical covariates" (blue) vs "Clinical covariates" (pink). **d** Venn diagram showing the number of correctly predicted relapse outcomes specific to or common between "Cam_121"(blue) and "clinical covariates" alone (pink). Out of a total of 143 patients, 30 were wrongly predicted by both the models in **c** and 36 were wrongly predicted by both the models in **d**. **e** Plot showing the performance of different classification models in predicting metastases in terms of AUROC (mean ± standard deviation) when trained on the AVAST-M lymph node dataset (*n* = 143). Within each panel, 14 different machine learning classifiers were trained to predict metastases: seven using tenfold CV (repeats = 1000) and seven using bootstrap resampling method (repeats = 1000). The two horizontal lines indicated within each panel denote the median AUROC of these seven classifiers, respectively. Statistical comparison using one-sided two-sample Welch *t* tests are indicated (bootstrap in green and tenfold CV in orange), see also Supplementary Tables S2c-d. Gerami_27, LMC_150: Leeds Melanoma Cohort 150 gene signature. See "Methods" section "Machine learning analysis" for details about the classification algorithms. Source data are provided as a Source data file.

We undertook further multivariate Cox regression analyses for all protein-coding genes in this dataset (*n* = 19,427) and found no single gene was significantly associated with either PFS or OS after correcting for multiple testing (*p* value < 0.05) (Supplementary Data 2 and Supplementary Data 3).

**A high weighted Cam_121 score reflected a lymphocyte depleted tumour with worse clinical outcomes.** In order to identify the biological processes reflected by the signature, we ran pre-ranked gene set enrichment analyses on genes ordered by their shrunken log-fold change from the covariate-corrected

differential expression analysis (see "Methods" section "Gene set enrichment analyses"). In doing this, we found that the top five significantly (FDR-corrected $p$ value <0.05) downregulated hallmark gene sets resulting from this analysis included interferon (IFN) gamma response, IFN alpha response, allograft rejection, inflammatory response and IL6-JAK-STAT3 signalling (Supplementary Fig. S13). Interestingly, when we ran gene set enrichment on DEGs derived from the (entirely separate) LN samples ($n = 143$; Supplementary Fig. S14), we observed significant (FDR-corrected $p$ value <0.01) downregulation of the exact same immune-related processes (Supplementary Fig. S14c). Therefore, indicating that the differential expression analyses (with a predominance of downregulated genes in both the primary melanoma and regional LN datasets) also reflect a significant downregulation of key immune-mediated processes in the samples from patients that developed metastases (Supplementary Figs. S3b, S13a and S14c).

We next used the Angelova dataset[25] to deconvolute the expression of immune cell subtypes within each sample (see "Methods" section "Immune cell correlation analysis"). We found a negative correlation between the weighted signature score and multiple immune cell types (with the highest correlation found with activated B cells ($\rho_{\text{Distant-Metastases=No}} = -0.8$, exact two-sided $p_{\text{Distant-Metastases=No}} < 2.2 \times 10^{-16}$; Fig. 4a), T cells ($\rho_{\text{Distant-Metastases=No}} = -0.73$, $p_{\text{Distant-Metastases=No}} < 2.2 \times 10^{-16}$; Fig. 4c), as well as the overall immune cell expression score ($\rho_{\text{Distant-Metastases=No}} = -0.75$, $p_{\text{Distant-Metastases=No}} < 2.2 \times 10^{-16}$; Fig. 4e). We also found that samples with a high weighted Cam_121 expression score were more likely to develop metastases than samples with a low weighted Cam_121 expression score (Fig. 4b, d, f and Supplementary Table S3). Although nine of the Cam_121 signature genes were common with the Angelova immune marker genes (*TUBB*, *AIM2*, *CASQ1*, *NTRK1*, *FASLG*, *CCR3*, *P2RY14*, *PRF1* and *CCR5*), we were able to demonstrate that samples segregated based on overall immune cell score, with low immune cell expression clustering with high weighted Cam_121 gene expression scores (Fig. 4g), using PCA.

We further explored the relationship between the weighted Cam_121 gene expression score and histopathologically assessed tumour-infiltrating lymphocyte (TIL) scores (applying independent scoring criteria with both the Clark[26] and Melanoma Institute Australia (MIA) scores[27], see "Methods" section "Tumour infiltrating lymphocyte analysis"). This further confirmed a significant negative correlation between the Cam_121 signature and TIL scores, such that a higher signature score equated to a relatively immune-deprived tumour with consequent worse clinical outcomes (Fig. 5). It is nonetheless important to point out that the Cam_121 signature retained its

prognostic influence even following correction for pathologically assessed TIL scores, and this remained consistent both within the AVAST-M and the external validation dataset from the LMC (Figs. 1c and 2e, respectively).

**Discussion.** The ability to identify primary melanoma patients at risk for disease recurrence is an important unmet need and effective prognostic biomarkers that could serve to guide adjuvant therapy are lacking. The 31-GEP assay (Gerami_27) has been developed in an attempt to address this clinical dilemma, however, the majority of published studies evaluating its performance have been retrospective or prospective cohort studies without a comparator group[28], and its use has not been advocated in established clinical guidelines[17]. We sought to identify whether the expression of genes in a primary melanoma tumour could predict for distant metastasis and survival, analysing data acquired from one of the largest phase III prospective adjuvant melanoma clinical trials associated with long-term patient outcome data[19]. We used covariate-corrected differential expression analyses to identify 121 genes significantly associated with distant metastases, which made up our signature, and found that this added prognostic value in both the prediction of metastasis and survival. The prognostic relevance was further confirmed in two independent external validation cohorts. Immune cell deconvolution analyses revealed that the weighted Cam_121 expression score negatively correlated with multiple measures of lymphocyte infiltration, with a high weighted signature score reflecting a relatively cold tumour immune microenvironment with worse long-term prognosis. These findings were cross-validated using pathologically assessed TIL scores, as well as gene set enrichment analyses, the latter showing that differential expression analyses in both primary melanoma ($n = 194$) and LN ($n = 143$) datasets reflected downregulation of the same key immune-mediated processes in association with metastases. That this conclusion was reached using unbiased differential expression, reaffirms the central importance of the immune system in this context.

The melanoma microenvironment consists of multiple immune and stromal cells, which play a critical role in regulating both the initiation and development of disease. Several studies have demonstrated the association of lymphocyte infiltration with longer survival[29–31], as well as an inverse relationship between TIL grade and the presence of LN metastases[27,32], implying that evaluating the tumour microenvironment landscape may hold promise for prognostic biomarkers. However, only a limited number of studies have investigated the immune landscape in primary melanomas. A transcriptomic analysis of primary melanomas identified six distinct subgroups based on their expression of immune-related, keratin and beta-catenin pathway genes[33]. In this study, patients with low immune but high beta-catenin score (CIC4) had the poorest OS[33]. A recent study utilising high-throughput sequencing of T-cell receptor beta-chain in T2–T4 primary melanomas ($n = 199$) indicated that the T-cell fraction accurately predicted PFS and was independent of other key clinico-pathologic covariates[34]. Although in our study it was difficult to discern specific immune cell subtypes using bulk RNA sequencing, given that the weighted Cam_121 score was strongly negatively correlated with B cells, T cells and all immune cells ($\rho_{\text{Distant-Metastases=No}} = -0.8$, $-0.73$ and $-0.75$, respectively, with exact two-sided $p_{\text{Distant-Metastases=No}} < 2.2 \times 10^{-16}$) and that IFN pathways dominated gene set enrichment, we regard this as further evidence that a successful immune-mediated cytotoxic anti-tumour response exists in primary melanoma. Critically, we found that the signature retained its prognostic power even after correcting for TIL score, and it is our opinion that quantifying the expression of these key immune-mediated genes could potentially provide a more standardised and reproducible measure of

**Table 1 Clinical utility of GEP test 5-year melanoma-specific survival (Leeds Melanoma Cohort Data).**

| AJCC stage | GEP_class | Death | Total | Proportion |
|---|---|---|---|---|
| 1 | All | 17 | 194 | 0.09 |
| 1 | High | 11 | 118 | 0.09 |
| 1 | Low | 6 | 76 | 0.08 |
| 2 | All | 76 | 279 | 0.27 |
| 2 | High | 64 | 192 | 0.33 |
| 2 | Low | 12 | 87 | 0.14 |
| 3 | All | 44 | 76 | 0.58 |
| 3 | High | 35 | 58 | 0.60 |
| 3 | Low | 9 | 18 | 0.50 |

Number and proportion of deaths per combination of AJCC stage and Cam_121 risk level estimates (based on 0.33 quantile cut off on weighted Cam_121 gene expression score expression of all 121 genes). Source data are provided as a Source data file.

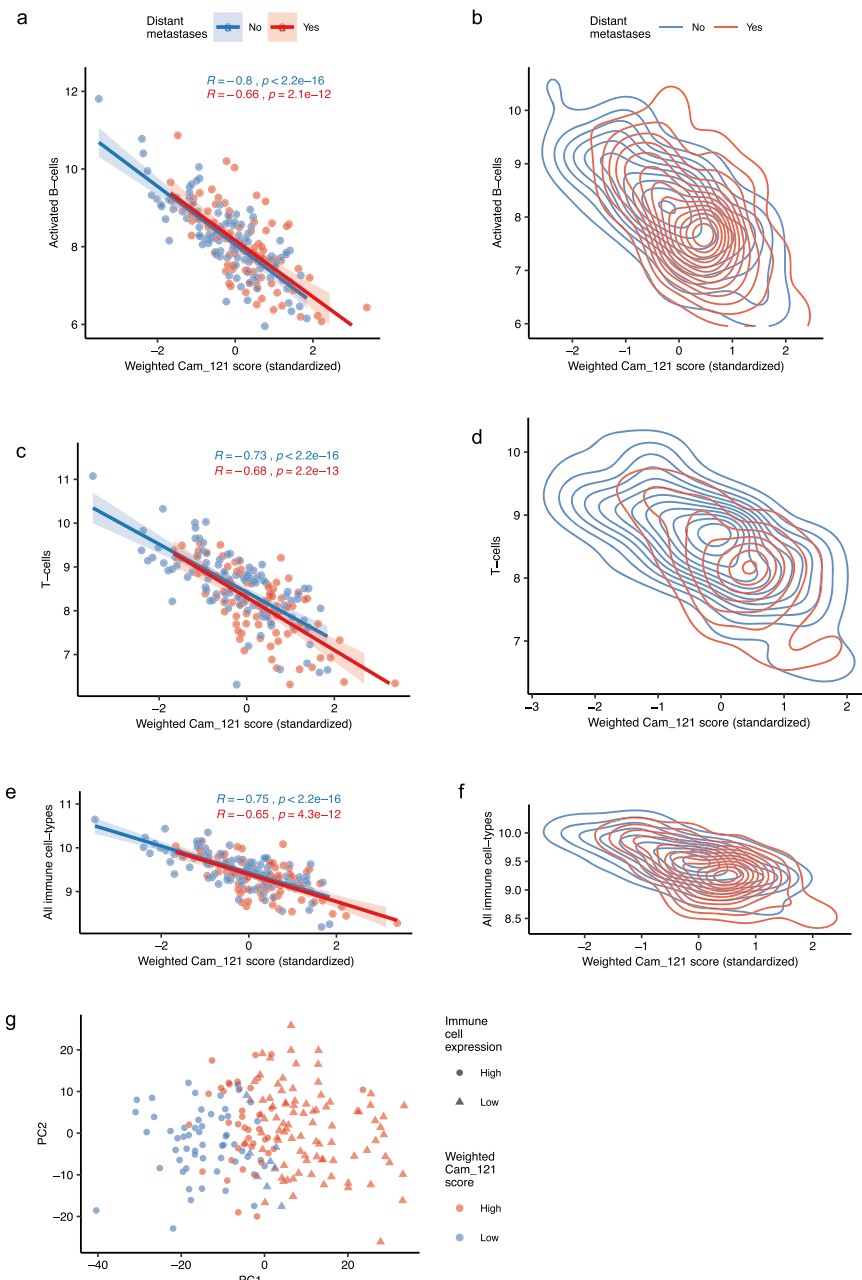

**Fig. 4 Weighted Cam_121 score negatively correlates with immune cell expression scores, indicating that a lower weighted signature expression score is associated with a richer immune microenvironment and better prognosis.** Scatterplots and density plots showing the relationship between the standardised weighted Cam_121 score (x-axes) and **a**, **b**, the median activated B-cell gene expression (y-axes), **c**, **d** the median T-cell gene expression (y-axes) and **e**, **f** the total immune score (median gene expression of all cell types derived from Angelova et al.[25], (y axes). Observations and lines of best fit are colour-coded according to their metastatic status, with red indicating relapse and blue indicating no relapse. The shaded region in the scatter plots of **a**, **c** and **e** corresponds to the 95% confidence interval of the line of best fit. The two-sided p values from the Pearson correlation coefficients are indicated for scatter plots. **g** Scatterplot of the scores of the observations on the two-first dimensions of a PCA analysis on the overall immune cell expression data. Observations are colour-coded according to their weighted Cam_121 expression score ("high"/"low" classification based on a quantile 0.33 split, indicated in red/yellow, respectively). Different symbols are used for observations with "high" (circles) and "low" (triangles) immune cell expression levels based on a median split. Source data are provided as a Source data file.

immune activity. Furthermore, the prognostic relevance in both primary melanoma and LN datasets attests to the signatures' robustness. The challenge over the coming years will be to identify and validate a clinically relevant measure of lymphocytic abundance of relevance to primary CM, that can be easily implemented in real-life clinical practice. These studies will also need to consider aspects of cost-effectiveness, which have not been explored in this analysis.

Interrogating the LMC, we found that the GEP-designated high/low risk could be used to separate patients with ≥33% risk of death at 5 years; a risk threshold for which we believe adjuvant systemic therapies could be considered. Conversely, it is envisaged that "low-risk" GEP profiles could also be used to "downstage" stage III patients for whom treatment might be unnecessary. There is substantial evidence supporting the importance of pre-treatment immune cell infiltration in eliciting

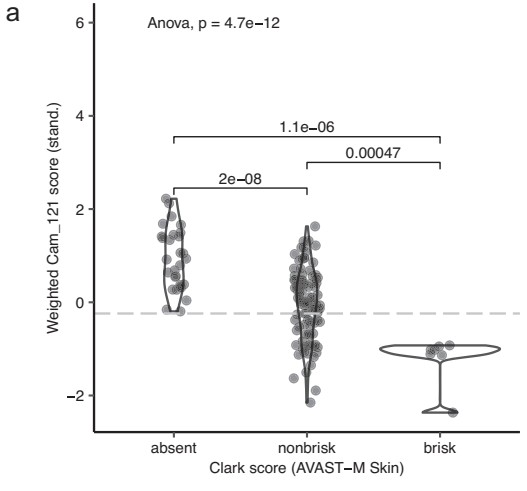
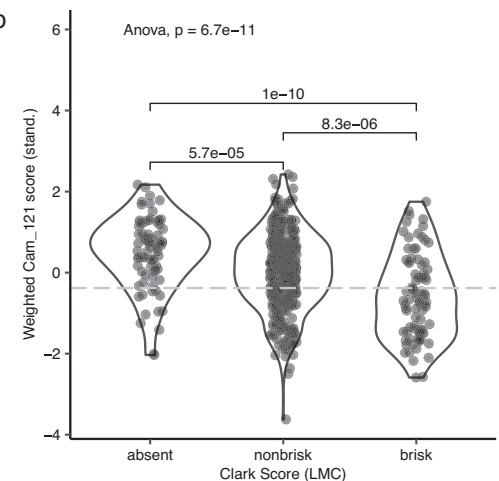
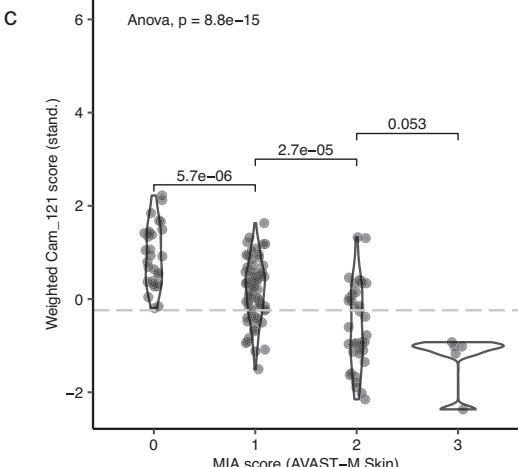

**Fig. 5 Cam_121 negatively correlates with tumour-infiltrating lymphocyte counts.** Violin plots showing the weighted (standardised) Cam_121 scores (y-axis) by levels of the TIL scores (x-axes) in the **a** AVAST-M primary melanoma dataset (n = 137) using Clark et al.[26] TIL scores, **b** Leeds Melanoma Cohort dataset (n = 499) using Clark et al. TIL scores and **c** AVAST-M primary melanoma dataset (n = 139) using MIA[27] TIL scores. The overall p values per plot are calculated using Fisher's ANOVA, and pairwise comparison p values were defined using two-sided Student's t test. The grey dotted line represents the 33% quantile cut off for the gene expression signature in each dataset. Source data are provided within the Source data file.

anti-tumour responses with immunotherapy[35], however, it remains to be seen whether Cam_121 expression can predict therapeutic responses in this setting. Future well-designed prospective clinical trials will ultimately be required to examine whether Cam_121 can be used to better tailor adjuvant therapy for early-stage melanoma patients.

The present study has a number of advantages over previous analyses. First, the large sample size linked to a well-conducted prospective clinical trial enabled an objective assessment of the risk of distant metastases, in addition to the key survival measures of interest. Furthermore, the long duration of follow up (minimum of 6 years) in a cohort of patients predating modern approved adjuvant systemic therapies provided a unique insight into the "natural history" of primary CM. Finally, to our knowledge, this is the first large-scale biomarker analysis in primary melanoma to make use of data from comprehensive RNA sequencing. That such unbiased genome-wide assessment uncovered the dominance of immune-mediated genes reaffirms the central role of the host immune system's ability to respond to the tumour, resulting in immune editing or in some immune control.

It is important to point out that high-quality evidence guiding the best practice use of gene expression predictors, particularly in the context of early-stage CM are lacking. Future trials evaluating adjuvant therapies should examine both primary and locoregional melanoma samples using full RNA-sequencing technologies, to better characterise a molecular subtype/signature that could ultimately be used in conjunction with existing CM staging parameters and tailor future interventions more specifically to the individual. We believe that measures of lymphocytic infiltration should also be assessed. Ultimately such studies will need to show that randomising early-stage melanoma patients based on a high-risk Cam_121 GEP to an intervention (or a change in surveillance) leads to improved outcomes[36].

Our results indicate that the Cam_121 signature score complements conventional melanoma staging by contributing prognostically relevant information and could potentially be used to select early-stage melanoma patients at higher risk of relapse or death. Further carefully designed prospective clinical trials will help guide how molecular features can be incorporated with traditional clinico-pathologic features to best estimate individual risk and guide the optimal clinical use of molecular biomarkers.

## Methods

**AVAST-M melanoma cohort**. This study made use of individual patient-level and transcriptomic data from the phase III adjuvant AVAST-M study, investigating the role of the angiogenesis inhibitor bevacizumab vs placebo in high-risk primary CM[18,19]. One thousand three hundred forty-three stage American Joint Committee on Cancer stage IIB (T3bN0M0 and T4aN0M0), IIC (T4bN0M0) and III (TxN1–3M0) cutaneous melanoma (seventh edition AJCC[20]) were recruited to the study over the period July 18, 2007–March 29, 2012, as previously described. The study (including the collection of DNA and RNA) was ethically approved in accordance with the Declaration of Helsinki (REC reference number 07/Q1606/15, 16th March 2007). Participants provided written informed consent to sampling of their tumour blocks during study recruitment (and prior to the investigational systemic therapy).

All study participants underwent a sentinel LN node biopsy, and if positive proceeded to a completion LN clearance as per the study protocol. Demographic (including gender, age, centre, as well as pathologic data (site of primary, Breslow depth, ulceration, LN involvement and *BRAF/NRAS* mutation by pyrosequencing)) was collected at the time of randomisation. Data were also collected on the timing, presence/absence and site/s of distant metastases (according to the findings from CT scanning). Data on overall and progression-free survival were collected with a minimum of 6 years follow up.

RNA-sequencing data was available on 204 primary melanoma samples of which 10 samples were removed from the downstream analyses owing to lack of data on all the clinical covariates, and 175 regional LN samples of which 32 samples removed from the downstream analyses owing to a lack of data on all clinical covariates (Supplementary Fig. S1).

**Leeds Melanoma Cohort**. A primary melanoma transcriptomic dataset from the LMC study (LMC DASL array) was used as independent replication. This represents a population-controlled cohort study, as previously described[16]. This study recorded data on MSS in 677 patients, calculated from the time of diagnosis to the time of last follow up or time of death from melanoma, whichever occurred first. The regression coefficient (beta) for each gene (reflecting differential expression in AVAST-M dataset) was used to generate a weighted signature score in the new dataset. Hence for further analysis, a per-sample weighted gene expression score for our Cam_121 gene signature was calculated by multiplying the expression value of each gene by its corresponding beta coefficient (Eq. 1) followed by z-score normalisation (zero mean–unit variance).

$$\text{Weighted signature score} = \sum_{i=1}^{n} \beta_i \times \text{gene}_i \qquad (1)$$

where *i* ranges from 1 to number of genes in the signature and $\beta$ corresponds to the beta coefficient of the respective gene obtained from DESeq2 analysis on AVAST-M melanoma cohort.

**Lund Melanoma Cohort**. Gene expression data on 223 primary tumours was generated using the Illumina DASL platform, as previously described[22]. Data on relapse-free survival as well as OS were collected. The DASL platform analysed 7752 genes and only 24 of the Cam_121 genes were present. Validation was undertaken using weighted signature scores as outlined above (Eq. 1).

**The Cancer Genome Atlas-SKCM Cohort**. The clinical and gene expression data from TCGA-SKCM[21], was downloaded from the cBioPortal[37]. The TCGA-SKCM dataset included only CMs, in particular any melanomas within these datasets from acral, mucosal and other rarer sites were excluded. Samples were filtered to a single sample per patient giving a total of 375 samples from 375 patients (including 87 primaries, 72 cutaneous relapses and 216 regional LNs).

**Australia Melanoma Genome Project Cohort**. All fresh frozen and FFPE samples were obtained in a method that was compliant with the relevant ethical regulations for work with human participants. The fresh-frozen tissue from the biospecimen bank of MIA[23]. All tissues and bloods form part of prospective collections of fresh-frozen samples accrued with written informed patient consent. The study was approved by the Sydney Local Health District RPAH zone ethics committee (Protocol No. X15-0454—previously X11-0289 and HREC/11/RPAH/444; Protocol No X17-0312—previously X11-0023 and HREC/11/RPAH/32; and Protocol No X15-0311—previously X10-0300 and HREC/10/RPAH/530). All samples were independently reviewed by expert melanoma pathologists to confirm the origin of each tumour from cutaneous skin.

Total RNA was extracted from fresh-frozen tissue using a mirVana miRNA Isolation Kit (Applied Biosystems, AM1560). RNA quality and presence of a small RNA fraction were measured using the Agilent 2100 RNA 6000 Nano and small RNA kits. RNA sequencing was performed using 1 μg of total RNA, which was converted into messenger RNA libraries using an Illumina mRNA TruSeq kit. RNA sequencing was performed using 2 × 75 bp paired-end reads on an Illumina Hiseq2000. Small RNA sequencing was performed using 1 μg of total RNA, which was converted into a small RNA libraries, size selection range 145–160 bp (RNA of 18–33 nucleotides) using Illumina's TruSeq Small RNA Library Preparation Kit and sequenced on an Illumina Hiseq2000 using 50 bp single-read sequencing with

1% control spiked in. RNA sequence reads were aligned to transcripts corresponding to ensemble 70 annotations using RSEM, raw sequences are available under study accession EGAS00001001552. Data from 55 samples from a mixture of tissue sites (including primary tumours, regional LNs, distant metastases, in-transit metastases and others) were available for this analysis.

**mRNA extraction**. Histopathological assessments of hematoxylin and eosin (H&E) stained slides were used to facilitate tumour sampling. Samples were consistently extracted from the least inflamed, least stromal regions of the invasive front of the tumour. RNA was extracted using the Roche High Pure FFPE RNA Micro Kit (cat# 04823125001; Genentech Biosciences) according to the manufacturer's recommendations. RNA quantity and quality were assessed using Agilent's 2100 bioanalyzer.

**Expression data generation**. Extracted RNA was sequenced on the Illumina exome-capture sequencing platform, using 50 base-pair paired-end sequencing. Quality control (QC) was performed using fastq_utils (https://github.com/nunofonseca/fastq_utils; v0.14.7) and FastQC (http://www.bioinformatics.babraham.ac.uk/projects/fastqc/; v0.11.7). The reads that passed QC were aligned to the reference genome (GRCh38) using TopHat2 (ref. [38]). Aligned reads were quantified using HTSeq[39]. Only those genes with more than five reads, as reported by HTSeq, in at least one sample were selected for further analysis. The sequencing data was of good quality with a median of ~50 million read-pairs/sample. A total of 446 tumour transcriptomes were profiled which included samples from: primary tumour (*n* = 204); LN (*n* = 175); local/distant relapse (*n* = 58) and uncategorised samples (*n* = 9). However, due to the clinical value of primary tumours in facilitating stratification at the earliest disease timepoint, we chose to focus our analyses on samples from cutaneous primaries (*n* = 204) and used the LN samples as an internal validation.

**Clinical covariate selection**. Firstly, the association between distant relapse (yes/no) and clinical covariates was studied both with or without controlling for length of follow up (defined as the time from diagnosis to last follow up) and for treatment (yes/no). When ignoring length of follow up and treatment, generalised Cochran-Mantel-Haenszel tests (R-package coin[40] v1.3-1) were used for nominal clinical predictors as they have the Pearson's Chi-square tests and Cochran-Armitage trend tests as special cases, when respectively considering the clinical covariate of interest as categorical or ordinal. For ordinal clinical covariates, we reported the "nominal/nominal" association results when the "nominal/ordinal" one was found less significant (as it is likely a sign that the assumption of linearity required by the ordinal test was not met). Mann–Whitney–Wilcoxon tests were used for continuous clinical covariates. When controlling for length of follow up and treatment, likelihood ratio tests comparing the fits of logistic regression models with and without the clinical predictor of interest were used. The *p* values were corrected for multiple testing using the Holm–Bonferroni method (Supplementary Table S1a). Note that, as the five-level stage variable was highly related to Nclass (Spearman correlation coefficient over 0.85), we picked the one with the lowest number of levels.

The variables two-level Breslow staging and two-level ECOG were significantly associated with relapse. The variable two-level treatment was found to be related to relapse, but is kept as control and the two-level EventMet was the variable of interest indicating whether the patient relapsed or not. Therefore, the covariates Stage, Breslow thickness, ECOG and treatment were accounted for in the design formula of DESeq2 (ref. [41]) without interactions and for further downstream analysis.

Secondly, the association between clinical covariates and OS (calculated from the time of diagnosis to the time of last follow up or death) and PFS (calculated from the time of diagnosis to the time of last follow up or death/progression to metastatic disease, whichever occurred first) was assessed by means of Cox proportional hazards models (R-package survival[42]). Both outcomes were considered as left-truncated due to delayed patient enrolment and right-censored due to loss of follow up or alive at the time of the end of the study. Six years was chosen as the minimum cut off for these analyses based on the original trial design.

Stage, sex, age and Nclass were significantly associated with both OS and PFS (*p* < 0.05; Supplementary Table S1b, c). The state of distant relapse ("EventMet") was the most important variable but was not of our interest, hence dropped. ECOG is a good predictor. Treatment was not significant (*p* > 0.05), but was kept in the analysis. Also for PFS, ulceration (Ulc) was borderline at 5% level and was dropped from further analysis. Therefore, stage, sex, age, Nclass, ECOG and treatment were corrected for in subsequent gene-level survival analyses. The AVAST-M primary melanoma dataset was also corrected for TIL counts (Clark Score, see also section "Tumour-infiltrating lymphocyte analysis").

**Differential expression analysis**. Differential expression analysis between primary tumours that became metastatic vs those that remained non-metastatic over the 6-year study period was performed using the package DESeq2 (ref. [41]; v1.18; R v3.6.1). The negative binomial models we considered controlled for the clinical covariates Stage, Breslow thickness, ECOG and treatment, as well as for the library size (offset). Raw read counts were provided as the input, with each column

representing a sample and each row representing a gene, along with the categorical clinical information about each sample as colData. Samples with missing information for any of these four covariates were removed from the analysis, leaving 194 samples in total. The adjusted $p$ value cut off (FDR) was set to 0.1 using the alpha parameter in DESeq2 results function and genes with FDR <0.1 were considered differentially expressed.

Log-fold change shrinkage was applied using the lfcshrink function with apeglm method from the apeglm package[43] (v1.6.0; R v3.6). For visualisation and other downstream analysis, variance stabilising transformation (vst) was used by means of the DESeq2's varianceStabilizingTransformation function with option blind = FALSE.

**Machine learning analysis.** This section explains the steps followed to develop a ML classifier for each signature and to evaluate their performance in predicting relapse (yes/no). A summary of the pipeline is outlined in Supplementary Fig. S15. The following steps were conducted using the following packages; caret[44] v6.0-86, DESeq2 v1.28.1; R v4.0.2, snakemake[45] v5.17.0. The AVAST-M primary melanoma dataset ($n = 194$) was used for classification model development and the AVAST-M LN dataset ($n = 143$) was used to test the performance of the final model (internal validation). In an independent analysis, we used only the AVAST-M LN dataset for both training/testing ("Methods" section "Model development and selection").

*Dataset preparation and pre-processing.* The AVAST-M primary melanoma (training) dataset was prepared such that each column/feature contained information about all the patients/samples. These features could either be the expression values of the genes within the signature and/or the categorical clinical covariate metadata. In the latter case, the categorical clinical covariates were converted to numeric dummy variables using one-hot encoding. The clinical outcome data (relapse vs non-relapse) were used as labels for the training. The AVAST-M LN dataset (testing) was pre-processed in the same way, such that the order of the features was preserved as in the training dataset and in case where the clinical covariates were used as features, they were converted to dummy variables from the train clinical covariates using the predict function in R.

In case of the gene expression data, vst transformation was applied to both training and testing dataset. To apply vst transformation on the testing dataset, mean-dispersion estimates learnt on the training dataset were used.

Next, the features corresponding to near-zero variance were removed using the default parameters of the trainControl function (R-package caret; i.e., freqCut = 95/5, uniqueCut = 10). The same feature(s) were removed from the testing dataset before evaluating the performance of the fully trained model.

*Model development and selection.* The aim of this analysis was to critically assess whether the Cam_121 gene expression signature (with or without clinical covariates as features) could outperform clinical covariates alone in predicting relapse (yes/no). We also compared this to the predictive power of two published prognostic signatures (LMC_150 (ref. [16]) and Gerami_27[12]) in an independent analysis. In this model, the training/testing was carried out on the AVAST-M LN dataset instead of AVAST-M primary melanoma dataset on which feature selection was performed for Cam_121 and clinical covariates. This was undertaken to reduce the risk of over-optimistic results that might arise from feature selection and testing from the same dataset. In developing a classification model for each of these five signatures of interest, seven different ML classifiers were considered, including; Bayesian generalised linear model[46] (bayesglm), Lasso and elastic-net regularised generalised linear model[47] (glmnet), k-nearest neighbour[48] (knn), linear discriminant analysis[49] (lda), random forest[50] (rf) and support vector machine[51] with linear (svmLinear) and radial kernel (svmRadial).

To avoid overfitting, repeated tenfold cross validation (repeats = 1000) was performed for model development and evaluation. Leave-one-out cross validation method (Supplementary Fig. S15) and bootstrap resampling method (Fig. 3e; repeats = 1000) were also tested to see if the choice of resampling method altered our results. This was implemented using the trainControl function (R-package caret).

At each training step, a random search was performed using 100 random (combinations of) hyperparameter(s) and the set of hyperparameter(s) leading to the largest maximum AUROC estimate on the training dataset was selected. Using this approach, we obtained 14 different classification models (7 classifiers × 2 resampling methods) for each of the five different signatures. To select the final best performing classification model for each signature, the model giving the highest AUROC value was selected for testing on the AVAST-M LN dataset.

When training the random forest classifier on the feature set Cam_121 with clinical covariates, we estimated the average feature importance score based on repeated tenfold CV and displayed them by means of boxplots using geom_boxplot function in R.

*Gene expression vs covariate performance comparison per patient.* In order to compare the performance of the signatures: "Cam_121 + clinical covariates", "Cam_121" and "clinical covariates" on the AVAST-M LN dataset ($n = 143$) on a per-patient basis, the labels predicted by their respective best performing classifiers were extracted (tenfold CV + svmRadial, tenfold CV + svmRadial and tenfold CV

+ svmLinear respectively). Venn diagrams obtained by means of the venn.diagram function from the VennDiagram package[52] (v1.6.20) in R were used to visualise overlaps.

*Statistical analyses.* To check whether the Cam_121 gene expression signature performed better in predicting relapse (yes/no) than models built on clinical covariates alone across multiple ($n = 7$) classifiers, Welch Two Sample $t$ tests were used (R function t.test with option var.equal = FALSE, paired = FALSE and alternative = "greater") for each performance metric and each combination of signature and clinical covariate at the 5% level. The null hypothesis was that the true difference in mean performance across seven classifiers between both conditions ("clinical covariates alone" vs "signature with/without clinical covariates") equals 0, while the alternative hypothesis was that the true difference in means is >0.

To compare the AUROC obtained on the testing dataset, DeLong's tests[53] were used using the roc.test function with alternative "greater" from the pROC package[52] v1.16.2 in R, where the null hypothesis is two AUROC obtained from the model trained on gene expression as features and the model trained on clinical covariates as features are equal, while the alternate hypothesis is that the model trained on gene expression as features performs better than the model trained on clinical covariates as features. The $p$ values and the $z$ decision threshold values from the test were reported.

**Determination of the weighted expression score cut-off to define "high" and "low" absolute risk of death at 5 years.** Data from the LMC were used to calculate the absolute risk of death at 5 years (this dataset was chosen for this analysis due to the preponderance of early-stage patients; stage I = 194 samples; stage II = 279 samples; stage III = 76 samples). Five-year MSS was calculated such that those patients who died due to melanoma within 5 years of follow up were assigned event = "Yes" and those that did not were assigned event = "No". Those patients who did not yet die and were followed up for <5 years were removed from the analysis due to inadequate follow up.

The quantile cut offs 0.25, 0.33 and 0.5 were used to divide patients into high/low groups based on their corresponding weighted Cam_121 expression score. Absolute risk of death at 5 years was calculated as the ratio of patients where event = "Yes" to the total number of patients within each stage (I–III). The cut off giving the maximal separation (of absolute risk of death) between high/low groups was selected. This was achieved using a 0.33 quantile cut off of the weighed Cam_121 expression score and subsequent references to high/low Cam_121 risk groups refer to these high/low weighted stratification cohorts.

**Survival analyses.** For each sample, a vector of weighted signature expression scores was calculated by using Eq. 1 on the vst normalised gene expression data. The standardised scores were then used as a continuous predictor in Cox regression models fitted by means of the coxph function of the survival package[42] (v3.1-12) in R (v3.6.3). The HR (95% CI) and $p$ values corresponding to the signature were reported in both univariate and multivariate analyses. Note that in case of the two published signatures, median gene expression scores were used instead of the weighted gene expression scores.

In order to display Kaplan–Meier (K–M) survival curves, samples were divided into "high"/"low" signature expression groups based on the 0.33 quantile cut off which we obtained from the absolute 5-year risk assessment in "Methods" section "Determination of the weighted expression score cut off to define "high" and "low" absolute risk of death at 5 years". Samples with weighted signature expression score greater than this cut off were assigned to the "high" group and those with weighted signature expression score lower than this cut off were assigned to the "low" group. Of note, we found that the 0.33 quantile cut off of the weighted gene expression scores were remarkably consistent across all the external datasets (data not shown). The survival distribution of both groups was finally compared by means of logrank tests using survfit function from the survival package (v3.1-12) in R (v3.6.3). The parallel processing was conducted using snakemake[45] v5.17.0. The K–M curves were plotted using plot function from the R-package graphics[54] v3.6.3 and ggsurvplot function from R-package survminer[55] v0.4.7.

**Tumour immune microenvironment analysis.** Sample-level gene expression data from the AVAST-M primary melanoma cohort was deconvoluted into infiltrating immune cell scores using the Angelova dataset[25]. This dataset reports 812 marker genes corresponding to 31 immune cell subtypes. Out of these 812 genes, 53 genes were missing from our 38,690 gene list. Therefore, two immune cell subtypes MDSC (myeloid-derived suppressor cells) and NK56_bright (natural-killer CD56bright cells), with >1% of missing marker genes were removed from further analysis, leaving 719 marker genes corresponding to 29 cell types.

*Immune cell correlation analysis.* To perform correlation analysis for each cell type, the median of the corresponding marker genes' expression for each sample ($y$-axis) was plotted against the weighted Cam_121 expression score for that sample ($x$-axis). To calculate the overall immune score, the median of all 719 marker genes was used for the analysis.

A regression line was fitted through these points using the geom_smooth function (method = "lm") of the ggplot2 package[56] (v3.3.0) in R (v3.6.2). The

Pearson's correlation coefficient ($\rho$) between the immune cell score and the signature, as well as the $p$ value of the corresponding test of association were estimated by means of the function stat_cor of the ggpubr package[57] (v0.2.5) in R (v3.6.2). Samples were coloured according to their metastatic status. With red points indicating the samples obtained from patients who later metastasised (Yes) and blue points indicating the samples obtained from patients who didn't (No). The density plots were made using the geom_density_2d function of R-package ggplot2 (ref. [51]; v 3.3.0).

To confirm the relationship between the signature and immune cells, a PCA analysis was performed. For each cell type, the corresponding marker genes' expression was projected into the principal component space and the first two principal components explaining the maximum variance were plotted against each other. These samples were then coloured by the "high"/"low" weighted Cam_121 expression score groups obtained during the survival analysis and shaped by the "high"/"low" groups based on marker genes' median expression corresponding to that particular cell type. Here, to divide the samples into two independent categories based on their marker genes' expression, the median cut off was used, where the sample with marker genes' median expression value above its overall median value was assigned to the "high" group and that with marker genes' median expression value below its overall median value was assigned to the "low" group.

*Gene set enrichment analyses.* Preranked GSEA (GSEA-P) was implemented using the GSEAPreranked tool of the GSEA software from Broad Institute[58,59] (v4.0.2). Hallmark gene sets were downloaded from the MSigDB database[60] (v6.2.0). The genes were preranked according to their shrinked log-fold change values obtained in the differential expression analysis and the GSEAPreranked tool was run with default parameters with the enrichment statistic set to "classic".

*Tumour-infiltrating lymphocyte analysis.* H&E slides corresponding to each of the 194 samples were digitally scanned to 40× magnification using the Vectra Polaris scanner from AKOYA biosciences. TIL scores were double blindly evaluated by two experienced pathologists. Two different scoring methods were used including; (i) the Clark scoring[26] and (ii) the MIA system[27]. This resulted in an agreement of 56% and 40%, respectively. This lack of consistency in scoring (particularly within the "non-brisk" group) has been previously noted in the literature[61,62]. We used a third independent expert pathologist to assess those slides where the two pathologists failed to agree. After removing the slides with poor scan quality, we had Clark TIL scores for 133 primary tumours and MIA TIL scores for 135 primary tumours.

Once the scores were obtained, a violin plot was plotted between the scores and the standardised weighted Cam_121 score using the geom_violin function of ggplot2 package. The $p$ values for pairwise comparisons were obtained using $t$ test and the global $p$ value was computed using ANOVA. This was implemented using the stat_compare_means function from the R-package ggpubr (v0.2.5)[57].

**Visualisation of inherent distribution of samples.** To visualise if the samples cluster by their metastatic status, PCA was performed on the primary melanoma samples ($n = 204$) and LN samples ($n = 175$) using 1000 most variable genes. This analysis was performed using the prcomp function from R-package stats[54] (v3.6.2), plotted using the qplot function from ggplot2 package[56] (v 3.2.1) and the scree plot was generated using the screeplot function, also using the R-package stats[54] (v3.6.2). The samples were further coloured by whether they metastasised or not and shaped by their tissue of origin.

**Testing signature performance against randomly selected genes.** In order to test the performance of our signature against randomly selected genes, random genes of the 121 gene length were selected from 19,434 protein-coding genes. The analysis was repeated using the exact same pipeline to compare its performance against our signature. This process was repeated 1000 times without replacement using 1000 different seeds and the $p$ value testing the significance of our signature was defined as a left-tailed event for predicting OS/PFS survival. This analysis was inspired from the SigCheck package[63] (v2.14.0) in R (v3.5.1), whereby the plotting function sigCheckPlotSurvival was modified to accept scores generated from our analysis.

**Power calculation for the external validation datasets.** To assess whether significant validation of our signature was likely in the external validation datasets, we performed a simulation-based power analysis considering $R = 2500$ Monte Carlo samples. Simulation parameters, like the proportion of events for both outcomes, the hazard ratios and the predictor and right-censored time-to-event distributions, were based on the AVAST-M study. The log-normal and exponential distributions were respectively chosen to model time-to-relapse and time-to-death from time-to-relapse. The normal distribution was selected to model censoring for both outcomes.

**Reporting summary.** Further information on research design is available in the Nature Research Reporting Summary linked to this article.

## Data availability
The raw RNA-sequencing data (forward and reverse fastq files) has been made available at the European Genome-Phenome Archive at the EBI under the following dataset accession ID: EGAD00001006401. The source data underlying this paper are also available through the GitHub repository: Manikgarg/MelanomaTranscriptomics [https://github.com/Manikgarg/MelanomaTranscriptomics/tree/master/Source_Data][64]. The clinical and gene expression data from The Cancer Genome Atlas (TCGA-SKCM), can be downloaded from the cBioPortal[37]. Data from the Leeds Melanoma Cohort[16], Lund Melanoma Cohort[22] and the Australia Melanoma Genome Project[23] are available from the source publications.The MSigDB database[60] gene set collections are available for download from http://www.gsea-msigdb.org/gsea/downloads.jsp#msigdb. Source data are provided with this paper.

## Code availability
The code to reproduce the results is available at the GitHub repository: Manikgarg/MelanomaTranscriptomics (https://github.com/Manikgarg/MelanomaTranscriptomics/tree/master/scripts)[64].

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

## Acknowledgements

We would like to thank Neera Maroo, Leticia Campo and the Translational Histopathology Laboratory at the Oxford Department of Oncology for sample biobanking and processing, as well as digitally scanning the primary tumour H&E slides. Vivek Iyer for cross-checking the differential expression analyses. Wolfgang Huber and Janet M. Thornton for advice on machine learning analyses. Michael I. Love for advice on data normalisation for machine learning. Nicholas K. Hayward for sharing RNA-sequencing data from the Australia Melanoma Genome Project. Danish Memon for expert input on immune cell deconvolution and Andrea Manrique-Rincon for her help on the critical interpretation of these analyses. Alastair Droop for critical review of the manuscript. This work was supported by Cancer Research UK and the Wellcome Trust. Bevacizumab was supplied by Genentech pharmaceuticals. R.R. was partially supported by the 2019 Cycle for Survival's Equinox Innovation Award. N.A.F. was partially supported by the European Union's Horizon 2020 research and innovation programme under grant agreement no. 668981. The Leeds Melanoma Cohort research was carried out with research funding CR-UK; C588/A19167, C8216/A6129 and C588/A10721 and NIH; CA83115. M.R.M. is supported by the National Institute for Health Research (NIHR) Oxford Biomedical Research Centre. The views expressed are those of the authors and not necessarily those of the NHS, the NIHR or the Department of Health. R.A.S. supported by a National Health and Medical Research Council of Australia (NHMRC) Program Grant and Practitioner Fellowship. G.V.L. is supported by the University of Sydney Medical Foundation. Support from the Ainsworth Foundation, the Cameron Family, as well

as from colleagues at Melanoma Institute Australia and Royal Prince Alfred Hospital are also gratefully acknowledged.

## Author contributions

R.R., A.B. and D.J.A. conceived the project. MRM co-ordinated the sample biobanking. M.W. and Y.Y. co-ordinated the mRNA extraction and RNA sequencing from the AVAST-M trial. R.R. co-ordinated the sequencing and clinical data extraction. N.A.F. derived the RNA-seq counts from raw sequencing reads. R.R., D.J.A., A.B. and M.G. developed the clinical questions and experiments and M.G. carried out the analysis. D.L.C. ran the statistical analyses. J.N., T.B. and J.N.B. validated the findings in the Leeds Melanoma Cohort. M.L. and G.B.J. validated the findings in the Lund Melanoma Cohort. P.C., M.R.M. and C.P. provided senior input on the translational scope of the project. S.M., N.S. and J.T. annotated histopathological slides with the tumour-infiltrating scores. I.V., S.L., F.N., J.S.W., J.F.T., G.V.L. and R.A.S. facilitated validation within the Australian Melanoma Genome Project. M.G. and R.R. wrote the manuscript. A.B. and D.J.A. provided overall study supervision. All authors approved the final manuscript.

## Competing interests

P.C. is an advisory board member at Roche. M.W./Y.Y. are employees and stockholders at Roche/Genentech. D.J.A. is a consultant at Microbiotica. R.A.S. has received fees for professional services from Qbiotics, Novartis, MSD Sharp & Dohme, NeraCare, AMGEN Inc., Bristol-Myers Squibb, Myriad Genetics, GlaxoSmithKline. G.V.L. is consultant advisor for Aduro Biotech Inc, Amgen Inc, Array Biopharma inc, Boehringer Ingelheim International GmbH, Bristol-Myers Squibb, Highlight Therapeutics S.L., Merck Sharpe & Dohme, Novartis Pharma AG, QBiotics Group Limited, Regeneron Pharmaceuticals Inc, SkylineDX B.V. J.F.T. has received honoraria for advisory board participation from BMS Australia, MSD Australia, GSK and Provectus Inc, and travel support from GlaxoSmithKline and Provectus Inc. All other authors report no conflicts of interest.
