## [Peer Review File · Nature Communications]

Reviewers' Comments:

Reviewer #1:

Remarks to the Author:

In this manuscript, Garg et al. describe their analysis of gene expression in 446 melanoma samples and their derivation of a 121-gene signature that predicts outcome in the long term. They use data from an adjuvant trial with long-term followup to generate their results and then validate their findings in a couple of other datasets, primarily the Leeds dataset. The manuscript is clearly written, the figures lucid and representative and the results are interesting.

Although the 121-gene signature is convincing and well-supported, several questions remain. First, upon re-analysis of the signatures presented by Thakur (2019) and Gerami (2015), would the authors independently find the 121 gene signature? How extensive are the overlaps in these three signatures? Second, can they validate their signature using the Cancer Genome Atlas data? Third, the results seem to indicate that tumor-infiltrating lymphocytes represent a positive prognostic factor for melanoma which is not an entirely new observation. Have they used archival paraffin collections to validate their findings using histological analysis to estimate tumor-infiltrating lymphocytes instead of using the more expensive and laborious RNA-seq method?

Minor comments:

Top of page 10: Is „inputted“ really a verb?

Reviewer #2:

Remarks to the Author:

This study identified a gene signature that can predict long-term survival outcomes of patients with Melanoma from the transcriptome of a cohort where patients are dichotomized into two groups with either good or poor prognosis. Taking this gene signature and several clinical covariates, the authors build and select the best performing machine learning model with fine-tuned hyperparameters. This new classifier showed better performance when compared to existing melanoma signatures and machine learning models with clinical covariates only. They further validated this signature other independent datasets. Overall, this study is important for the field of precision medicine and could lead to an clinical relevant signature for patient stratification that will reduce overtreatment and medical cost. However, there are several major concerns for the validity of the findings:

1. The 121-gene signatures were selected by using the whole dataset instead of leaving the validation sample out in each round of LOOCV. But feature selection should only be part of the training and not have validation set included. This is a major FLAW in training and testing supervised machine learning models. The selection of clinical variates seems to be dependent on the entire dataset as well, which again biased the selection results. LOOCV could be overly optimistic about the true accuracy. K-fold cross-validation should be used instead.
2. Does comparing the prediction performance on training dataset really support this idea? Since the model is trained on this dataset, there could be overfitting on the dataset but not performs equally well on any other external dataset. In addition, in the following validation on two external datasets, there seems to be no data showing the performance of classification but only survival analysis. The improvement over other methods on the two validation set is marginal. Is this due to hyperparam tuning? Are the other models tuned properly?
3. The author didn't show that the outperformance of gene signature over a model using only clinical variables is worth the extra cost for doing an assay for patients in practice though it's statistically significant.
4. Criterion used to determine high/low signature expression: According to the method part, the

cutoff for each dataset is the 0.33 quantile. It is a relative cutoff but not an absolute one. In practice, we can either assign a new patient its quantile in the training dataset or compare it with an absolute value. But during validation in these two external datasets, they are using 0.33 quantile cutoff in the new dataset, which could lead to a biased result.

5. The evaluation of clinical covariates in prediction power: the two groups of patients are classified based on the metastasis status while controlling for a set of clinical variables; thus it doesn't reflect the real prediction power of clinical covariates between these two groups after controlling them. The concern is whether the clinical covariates really has a poor performance. The author may need to explain more on how they control the clinical variables.

6. The result from Lund primary melanoma cohort (the second external dataset) is not convincing: as the author mentioned that only 24 of 121 genes were included in this dataset, it is not able to demonstrate the prediction ability of the original Cam_121. What is the performance of these 24 genes on the other 2 datasets? Is LMC_150 completely included in all three datasets?

7. The comparison between Cam_121 and random 121 genes: the author uses a p-value to compare the performance of different features. However, since both OS and PFS are binary predictions, It is more common to compare the performance of different features by the AUROC.

8. Paragraph 4 of section 6.2 is very confusing. It reads like they trained the model using nested-LOOCV for each classifier using each signature. However, according to the context, the models were trained using all the selected signatures and the clinical covariates.

Minor editorial issues like typos or format issues:

-Page 19-21, wrong item number 6.x after 8.

-Page 44 : The p-values are defined as the fraction of scores of random signatures which are greater THAN the one observed when using the (real) Cam_121 gene signature.

REVIEWER COMMENTS

Reviewer #1 (Remarks to the Author):

In this manuscript, Garg et al. describe their analysis of gene expression in 446 melanoma samples and their derivation of a 121-gene signature that predicts outcome in the long term. They use data from an adjuvant trial with long-term followup to generate their results and then validate their findings in a couple of other datasets, primarily the Leeds dataset. The manuscript is clearly written, the figures lucid and representative and the results are interesting.

We thank the reviewer for kindly reviewing our work, for highlighting the critical importance of this question and for the accurate summary of our study.

- 1. Although the 121-gene signature is convincing and well-supported, several questions remain. First, upon re-analysis of the signatures presented by Thakur (2019) and Gerami (2015), would the authors independently find the 121 gene signature?**

We thank the reviewer for this comment and question, which we have fully considered. In summary, due to differences in trial design and primary outcomes captured by these studies (point 1 below), as well as marked differences in sequencing platforms and downstream gene expression analyses (point 2 below), we are unable to apply the same methodology (covariate-corrected differential expression analyses comparing metastatic vs non-metastatic outcomes) to independently identify our 121 gene signature within these datasets. In response to these important comments made by the reviewer, we have now provided a more detailed description of the two aforementioned studies within the background. We have also added a further sentence highlighting the heterogeneity across these studies, which may therefore limit our ability to make direct comparisons between these studies (page 5).

We start with a short summary of the three studies:

Thakur 2019: The signature from Thakur 2019 (PMID: 31515461) was generated from analyses within the Leeds Primary Melanoma Cohort (LMC, n=687), a population-ascertained cohort study (PMID's: 19770375 & 25403087). In their study, mRNA extracted from primary tumours was analysed on the Illumina DASL-array platform and then clustered using a consensus clustering approach based on partitioning around medoids (PAM). This generated six distinct molecular classes. The genes were ranked based on their p-values for their differential expression across the 6 LMC classes. The top 25 upregulated genes within each of the six LMC classes were retained as the final refined signature giving a total of 150 up-regulated genes (herein referred as LMC_150). This signature's prognostic capabilities were then evaluated in a Cox regression survival model evaluating melanoma-specific survival within the LMC.

Gerami 2015: The 27 gene signature described in Gerami 2015 (PMID: 25564571) was based on data from several microarray based sequencing studies (PMID's: 15492234, 17289871, 10952317, 15833814, 21410771, 19795447 and 16251803). The expression of genes from primary tumours in these studies (in both cutaneous and acral melanoma) were compared to metastatic tumours. Their analysis led to the selection of 54-genes with notable difference in

their expression profile between primaries and metastases. Of these 54 genes, 20 were prioritised based on chromosomal loci (PMID: 25564571). This panel also included an additional 5 gene targets which, according to the authors, was based on unpublished data from uveal melanomas. The final 2 gene targets were selected on the basis of previous mutational studies on *BAP1*, and probes for both the 3' and 5' prime ends of the *BAP1* transcript were included in the final gene set. The Gerami 2015 study applied this final 27-gene consensus signature to RNA isolated from 107 primary cutaneous melanomas, sequenced using RT-PCR, in order to evaluate its performance in predicting disease-free survival. Subsequent studies evaluated the performance of this signature in the context of retrospective clinical studies.

Garg et al 2020: In contrast to the studies above, our study considered data from a phase III randomized clinical trial (PMID: 30010756), whereby full RNA sequencing data (with a median of 50 million reads/sample and including data on 58,303 genes) was generated on 446 early stage melanomas. In view of the uniform follow-up and imaging protocols across all participants within this prospective study, we were able to separate our cohort into patients that experienced distant metastases and those that did-not, over the study follow-up (minimum 6 years), see Supplementary Figure S1. Our signature was generated by undertaking differential expression across these two clinical cohorts, whilst correcting for other clinical covariates that independently contributed to this outcome (see Methods section 9). Our signature was then evaluated for its ability to predict metastases using a machine learning model, as well as overall and progression free survival (Supplementary Figure S1 and S15), validated in both a separate internal (Figure 2A-C, Figure 3B) as well as external validation datasets in case of the survival analyses (Figure 2D-I).

In reference to the reviewers' question enquiring whether re-analysis of these external datasets could independently identify our signature, it is important to consider the following differences between these studies:

1. **difference in primary outcome:** The Leeds Melanoma Cohort and the studies reported by Gerami and colleagues were retrospective in nature and could only evaluate survival as the outcome of interest. In particular, due to marked variability in follow-up and management protocols across patients within these studies, these datasets do-not permit the reliable separation of patients into metastatic and non-metastatic outcomes. The researchers of Thakur 2019 confirmed in private communications that dichotomizing participants from their study in this way was not feasible and that all the outcome data (including that of subsequent publications from this cohort) report on melanoma-specific survival as the key outcome measure. We have not attempted to contact the investigators of the studies by Gerami and colleagues; however based on multiple subsequent publications using this signature, it appears that the same would apply. Please note also that the raw data from Gerami et al (which appears to be at least partially sponsored by a private company), are not publicly available.
2. **partial vs genome-wide expression data:** As discussed more thoroughly in the next question, the LMC, Gerami and Lund datasets were generated using Illumina DASL-microarray platform, whereas data from our study were generated using RNA sequencing approaches. This leads to differences in signal to noise ratios, moreover RNA-seq provides us with genome-wide unbiased expression data. These differences inevitably lead to differences in signatures that can be derived.

How extensive are the overlaps in these three signatures?

Figure R1 (below) shows the overlap between the three sets of signatures in terms of gene names. With the exception of a small overlap of 4 genes between LMC_150 and Decision-Dx Melanoma, we can note that the list of signature genes selected by each study are different. To explain this lack of overlap, we will first discuss the differences in the input expression data (points 1 and 2 below) and second show that these sets of genes are not independent (point 3 below) as we do see a correlation between these signatures in terms of their gene expression and the biological processes they capture (Figure R3, R4, R5 and R6). In light of the comments made by the reviewer, we have now highlighted this point within text, particularly the limited number of Cam_121 genes identified in the Lund DASL array dataset (p.8). We have also highlighted the need for further expression datasets in this field using more sensitive sequencing technologies (p.17).

Figure R1: Venn diagram detailing the overlapping genes in these three signatures. There were only 4 genes overlapping between Decision_Dx (Gerami 2015) and LMC_150 which were *GJA1*, *S100A9*, *AQP3* and *PPL*.

1. Differences between sequencing data:

In considering this overlap, it is critical to highlight the heterogeneity in the clinical and experimental approaches across these differing studies. In particular, it is important to recognize that the LMC, Gerami and Lund datasets were generated using an Illumina DASL-microarray platform, whereas data from our study was generated using RNA sequencing approaches (see Methods section 6). In personal communications with the researchers of Thakur 2019, the authors have expressed that their approach, which represented the best suited technology at the time, would no longer be considered an adequate platform particularly from a sensitivity standpoint. A notable difference when acquiring sequencing data using microarrays compared to RNAseq is that the former requires discarding a larger number of genes. These are predominantly genes that are not highly expressed even though they may be informative, as RNAseq data have shown. In support of this, when we compared the absolute expression of each of the three signatures using RNA sequencing data acquired from primary melanomas in our study (Figure R2 below), as well as RNA sequencing from primary melanomas within the Cancer Genome Genome Atlas (PMID: 26091043, Figure R2B below), indeed we find that the Cam_121 signature genes are expressed at low levels relative to both the LMC_150 as

well as the Gerami signature genes. As a further check, we also included all the genes covered by the Lund microarray platform (PMID: 22675174), which formed part of the external validation of our signature's performance). All these genes were expressed at much higher levels than our signature genes (Figure R2 below).

Figure R2: Violin plot comparing the median expression of genes from the three aforementioned signatures (as well as all the genes within the Lund_DASL array) across two datasets: (A) Primary melanomas (n=194) from the AVAST-M study (PMID: 30010756) and (B) Primary melanomas and regional cutaneous relapses (n=159) from The Cancer Genome Atlas (PMID: 26091043).

These notable differences between microarray and RNA sequencing data provide an explanation for the lack of overlap between sets of genes observed in Figure R1. Although Figure R2 demonstrates that genes from Cam_121 are relatively lowly expressed in primary melanomas in comparison to these other signatures, it is important to emphasize that this is a feature of the more sensitive sequencing platform employed, and does not in any way influence the clinical validity of the signature in this context. The clinical relevance of the Cam_121 gene signature has been explored on multiple levels within this study (including differential expression analyses, as well as across both internal (Figures 2A-C and 3B) and external validation datasets (Figure 2D-I). Thereby showing that the Cam_121 gene signature can still be picked up using these less sensitive sequencing platforms and that their influence on survival remains consistent across different datasets.

2. Other differences:

In addition to the expression assay platform, it is important to point out that there are a number of other differences between these three aforementioned datasets that could further contribute to the lack of overlap between signatures. Let us highlight, for example, (i) variable inclusion criteria, particularly varying disease stages (and in the case of Gerami different melanoma subtypes with the inclusion of uveal melanomas), (ii) varying clinical follow-up protocols, (iii) prospective vs retrospective nature of the studies, (iv) differing methods of RNA

extraction, (v) and finally varying approaches to generate the signature as well as the downstream methods by which the signatures' performance was evaluated.

3. Non-overlapping genes nevertheless showed some correlation in biological processes:

Notwithstanding this, the reviewers' insightful comment prompted us to look more carefully into whether these signatures could reveal some common biological insight. To this end, we explored the correlation of each gene in Cam_121 with genes in both LMC_150 and Decision-Dx melanoma in our dataset. This was achieved using the R function `cor.test()` to test the association between paired samples using Pearson's product moment correlation coefficient with null hypothesis indicating no correlation, i.e., $H_0: \rho(X,Y) = 0$ versus $H_1: \rho(X,Y) \neq 0$, where $\rho(X,Y)$ denote the Pearson's correlation between the expression of gene X (of Cam_121) and gene Y (of LMC_150). Only the correlations showing a p-value smaller than 0.05 were kept in the downstream analysis (Figure R3).

Figure R3: Plot comparing the percentage of genes in each signature with the corresponding absolute Pearson's correlation coefficient in a pair-wise correlation test. A) Percentage of Cam_121 genes (red) correlating with percentage of LMC_150 genes (blue). B) Percentage of Cam_121 genes (red) correlating with percentage of Decision-Dx Melanoma genes (blue). Of note, none of the 12 genes from LMC_150 correlated with 4 Cam_121 genes (with absolute correlation coefficient $\rho > 0.8$, p-val < 0.05).

In order to look further into these correlated genes, we chose a cut-off based on the strength of correlation between these genes, whereby only genes in the signature Cam_121 or LMC_150 with absolute correlation coefficient greater than 0.8 were considered. Interestingly, all 12 of the LMC_150 genes that correlated with 4 of the Cam_121 genes (with these criteria) were all positively correlated (Figure R4).

Figure R4: Tile plot showing Cam_121 (x-axis) and LMC_150 (y-axis) individual genes with $\text{abs}(\text{Correlation coefficient}) > 0.80$. Note the scale, indicating that all considered pairs of genes showed an absolute correlation estimate greater than 0.8.

In order to gain an insight into the underlying biological processes that might be represented by these positively correlated genes, we undertook gene ontology biological process (GO-BP) overlap analysis using the MSigDB database (<https://www.gsea-msigdb.org/gsea/msigdb/annotate.jsp>). The list of genes names was provided and the “BP: GO biological process” option was selected. GO-BP gene sets with FDR q-value < 0.05 are shown in Figure R5 below.

Gene/geneset overlap matrix

Figure R5: Tile plot indicating the gene sets represented by the correlated genes (gene sets with FDR-corrected p-value <0.05) are shown, the Cam_121/LMC_150 genes present within each gene set are also indicated.

Although gene ontology analyses have their limitations and it would certainly not be feasible to pinpoint specific biological processes, it was interesting to note that all these gene sets relate to immune activation and regulation of the immune response. This indicates that at least a subset of the Cam_121 and LMC_150 correlated genes highlight the importance of the immune response in this context, which as the reviewer has outlined, has long been recognized to be prognostically relevant in this context.

Based on the result above we partitioned Cam_121 and LMC_150 genes into three separate clusters:

- **Cluster1:** Correlated Cam_121 and LMC_150 genes ($r \geq 0.8$),
- **Cluster2:** Cam_121 genes not correlated with LMC_150 genes ($r < 0.8$),
- **Cluster3:** LMC_150 genes not correlated with Cam_121 ($r < 0.8$).

We then undertook a Pearson’s correlation coefficient (r) between the median expression of genes in each of these three cluster with the corresponding median expression value of the three molecular classes described in TCGA-SKCM (PMID: 26091043, immune, keratin and MITF low), see Figure R6 below. In doing this we saw that Cluster 1, represented by the correlated Cam_121 and LMC_150 genes, had the highest correlation with the TCGA-SKCM “Immune” cluster ($r = 0.94$, p-value < 0.001), further supporting the finding that these correlated genes highlight immune-related processes. In contrast the two other (non-correlated) clusters had a weak correlation with other TCGA clusters, and GO-BP analyses revealed other gene-sets possibly indicative of other biological processes at play (not shown). We can thereby conclude that although there is clearly important overlap in the expression pattern of some of the genes in these signatures including some shared biological processes, there are also a number of non-correlated genes that may reveal other, as yet unappreciated biological insights. It is therefore important to point out that there may be more than one signature of prognostic importance and that these not necessarily be represented by the same genes. Different signatures may reveal different underlying biological processes.

Figure R6: Tile plot showing the correlation of the three aforementioned correlated (cluster 1) and non-correlated (clusters 2 & 3) gene expression clusters with TCGA-SKCM biological clusters. In each tile, the corresponding Pearson’s correlation coefficient value is shown. The tiles are colored according to the p-values obtained from cor.test() function in R.

Second, can they validate their signature using the Cancer Genome Atlas data?

We thank the reviewer for this valuable suggestion which we have now fully incorporated in the manuscript. **We can confirm that our signature does indeed validate in the TCGA-SKCM data (including primaries and lymph nodes; Figures R10 and R11 respectively)** and, in light of these comments, we have updated these results within the MS (Results section 2, p.8; Figures 2F-I).

Before addressing our validation findings, we first asked to what extent TCGA is a suitable validation dataset (Figures R7 and R8).

(i) Quality assessment:

- **Missingness:** As TCGA was designed primarily for molecular studies, clinical data collection was secondary and a number of data fields were not required by the programme such that data related to survival was often incomplete due to missing observations from the clinical event, e.g. death, cancer recurrence. In addition, we found that data was also missing from other key clinical covariates required in the multivariate model (Figure R7).

Figure R7: Missing data for each clinical covariate in TCGA-SKCM data. The x-axis represents each patient and the red bar indicates absent data for that patient. The percentage of information available for that clinical covariate is shown on the y-axis.

- **Time to relapse:** Centres participating in TCGA project were required to provide baseline clinical details when samples were first accrued and further details after 1 year of follow-up where possible. For many samples, the follow-up time could be as short as 1 year. In light of the relatively short follow-up records and as the power of survival models like the Cox regression mainly depends on the number of events, it might be argued that progression free survival (PFS) could be considered a more robust clinical endpoint than overall survival (OS), as patients normally develop disease recurrence before death and, therefore, more endpoint events are recorded during the follow-up period. However, the annotation for time of recurrence was often unreliable in this dataset as shown below (Figure R8). The incomplete annotation of clinical outcome data

in TCGA, together with short-term clinical follow-up intervals has already been noted by the research community (e.g. PMID's: 25109877 & 28472234). A further limitation of the TCGA survival data relates to the fact that samples were accrued both retrospectively and prospectively. One important consequence is that follow-up data were not collected uniformly and patients were followed up according to local clinic schedules rather than under a TCGA-specified protocol.

Figure R8: The relationship between time to relapse (x-axis) and time to death (y-axis). As expected, the time of death often occurred after the time of relapse, however in many cases both times matched suggesting that in these case either the time of relapse was arbitrarily set to the time of death or that the time of relapse matches the last follow-up time for those patients who were still alive.

In light of these limitations, survival studies using data from TCGA have recommended assessing survival from primary tumors, due to more complete clinical data collected at the time of initial diagnosis relative to metastases (PMID: 29625055). However skin cutaneous melanoma (SKCM) is unique among the TCGA tumor types because, among 472 tumors, only 87 are primary tumors, whereas 219 represented regional lymph node metastases, and 65 represented distant metastases. This is in contrast to other TCGA cancer types for which relatively fewer metastatic tumors were collected.

(ii) Power analysis: In order to define the sample size required to validate our signature in an external validation dataset (and whether 87 primary tumours from TCGA as above was sufficient in size) based on the effect size observed in the AVAST-M study, we performed a simulation-based power analysis considering $R=2500$ Monte Carlo samples. Figure R9 shows the power (y-axis) as a function of the sample size (x-axis) when modelling two survival outcomes, the time to relapse (green lines) and time to death from any cause (red lines), as a function of our signature, considered as a dichotomous (solid line) or continuous (dashed line) predictor, and when assuming characteristics noted in the AVAST-M study (proportion of events for both outcomes; distribution of the predictors, times to event [Log-normal for time to relapse and exponential for time to death] and right-censoring [normal for both outcomes]).

We can note that

- as expected, the power increases with sample size,
- for a given sample size, the power is larger
 - when considering the signature on a continuous scale,
 - when considering “time to death” compared to “time to relapse”

- a sample size greater than n=100 would be required to obtain a power greater than 80% in all cases.

Figure R9: Power (y-axis) as a function of the sample size (x-axis) when modelling two survival outcomes, the time to relapse (green lines) and time to death from any cause (red lines), as a function of our signature, considered as a dichotomous (solid line) and continuous (dashed line) predictor.

Validation of our signature on the TCGA validation dataset:

In view of the sample size limitations outlined above, we decided to combine the primary melanoma dataset from TCGA-SKCM (n=87) with regional cutaneous or subcutaneous tumours (n=72), giving a total of n=159 samples. Figure R10 shows the univariate results for overall survival on these samples (n=156; *continuous signature*: HR=1.5022, 95% CI = (1.151, 1.96), p=0.00273; *dichotomous signature (high)*: HR=1.925, 95% CI = (1.038, 3.57), p=0.0378). We further conducted a multivariate Cox regression model on these samples with adequate data on gender, age at initial pathologic diagnosis and pathologic stage, this showed (n=156; *continuous signature*: HR=1.591, 95% CI = (1.16525, 2.173), p=0.00348; *dichotomous signature (high)*: HR=2.432, 95% CI = (1.019, 5.802), p=0.0452). In light of these comments, we have now updated the manuscript with these analyses (Results section 2, p.8, Figures 2F-I).

Figure R10: Univariate Kaplan-Meier survival plot of high vs low risk Cam_121 signature groups in TCGA-SKCM primaries and regional cutaneous or subcutaneous melanoma samples (n=156). Expression cut-offs for the high/low risk groups were defined as the 0.33 quantile cut-off of the weighted Cam_121 gene expression score (standardized).

We further repeated the survival analysis with (n=216) Regional Lymph Node samples from the TCGA-SKCM dataset. In Figure R11 we show the univariate results for overall survival on all samples (n=212; *continuous signature*: HR=1.28454, 95% CI = (1.059, 1.557), p=0.0109; *dichotomous signature (high)*: HR=1.581, 95% CI = (1.005, 2.488), p=0.0476). We further conducted a multivariate Cox regression model on these samples with adequate data on gender, age at initial pathologic diagnosis and pathologic stage, this showed (n=212; *continuous signature*: HR=1.279937, 95% CI = (1.0306, 1.590), p=0.0256; *dichotomous signature (high)*: HR=1.486, 95% CI = (0.917, 2.406), p=0.1074). We have further updated the manuscript with these analyses (Results section 2, p.8, Figures 2F-G).

Figure R11: Univariate Kaplan-Meier survival plot of high vs low risk Cam_121 signature groups in TCGA-SKCM regional lymph node samples (n=212). Expression cut-offs for the high/low risk groups were defined as the 0.33 quantile cut-off of the weighted Cam_121 gene expression score (standardized).

Therefore we can conclude that despite the inherent limitations of survival analyses within this dataset, uni- and multivariate Cox regression analyses within TCGA-SKCM does indeed validate the prognostic power of the Cam_121 gene signature, and the results are entirely consistent with our original findings. In light of these comments, we have added a separate section within the beginnings of Results (section 2) focussed entirely on external validation of the Cam_121 gene signature within four external datasets including (p.7); '(i) the Leeds Melanoma Cohort (n=677)¹⁶; (ii) The Cancer Genome Atlas (TCGA-SKCM) (Skin = 159 and LN = 216); (iii) the Lund Primary Melanoma Cohort (n=223) the Australian Melanoma Genome Project; and (iv) the Lund Primary Melanoma Cohort (n=223).' The corresponding Figures have been consolidated in an updated Figure 2. We thank the reviewer for highlighting this important point which has undoubtedly helped strengthen the rigour and reproducibility of our study.

Third, the results seem to indicate that tumor-infiltrating lymphocytes represent a positive prognostic factor for melanoma which is not an entirely new observation. Have they used archival paraffin collections to validate their findings using histological analysis to estimate tumor-infiltrating lymphocytes instead of using the more expensive and laborious RNA-seq method?

We would like to thank the reviewer for this important and relevant point. Although data on tumour-infiltrating lymphocytes was not recorded within the original AVAST-M dataset, we agree that this is an important unanswered question and have sought out to address this point. We have therefore scanned all the available H&E slides for the primary tumours (n=194), which were then independently reviewed by three consultant dermatopathologists, each providing two separate tumour-infiltrating lymphocyte scores (Clark score and Melanoma Institute of Australia (MIA) score, see also Methods section 13.3). This process was unfortunately delayed due to lab closures and clinicians' time during the pandemic which ultimately delayed resubmission of this manuscript.

Overall, we find that the results were consistent with our original findings such that a high Cam_121 score associated with a lower TIL count (points 1 and 2 below, Figure R12). We have also updated the MS with this analysis, which now highlights that three independent analyses (morphological TIL count analyses, deconvolution of immune cell subtypes and gene-set enrichment analyses) all arrive at the same overall conclusion that; '*A high weighted Cam_121 score reflected a lymphocyte depleted tumour with worse clinical outcomes*' (Results section 7, page 14 and Figures 1C and 2E).

We would also thank the reviewer for highlighting the time and resource implications required for RNA sequencing technologies when compared to TIL count analyses. Firstly we should point out that rather than genome-wide RNA-seq (as employed by our study) future confirmatory clinical studies using the Cam_121 gene signature would only require targeted sequencing of the 121 genes, which is much cheaper. In addition, it is important to note that TIL counts from histopathological slides have their own inherent limitations, including subjectivity/lack of consistency, different scoring criteria as well as the need for highly-trained dermato-pathological assessment (see MS Method section 13.3). Our gene expression panel on the other hand not only gives an indication of the TIL counts but also helps to identify patients in early stage melanoma who may have worse outcomes and provides additional biological insights.

In response to the reviewer's important point, we have added a sentence in the discussion attesting to the need for cost-benefit analyses in future studies (p.15): '*The challenge over the coming years will be to identify and validate a clinically-relevant measure of lymphocytic abundance of relevance to primary CM, that can be easily implemented in real-life clinical practice. These studies will also need to consider aspects of cost-effectiveness, which have not been explored in this analysis.*'

1. Correlation of Cam_121 signature score with TIL count:

In the violin plot below we show the relationship of tumour-infiltrating lymphocyte score (x-axes) to the weighted Cam_121 signature score (y-axes) across both the AVAST-M and LMC (external validation) datasets, using both the Clark (Figure R12A+B) and MIA (Figure R12C) TIL scores. We found a significant negative correlation between the Cam_121 signature score and TIL score, such that a higher signature score conferring a worse clinical outcome and correlated with a poorer lymphocytic infiltration. This analysis validated the same finding we first identified using deconvolution analysis of immune cell subsets by gene expression (Figure 4), using an entirely separate approach.

Figure R12: The Cam_121 standardized gene expression score was significantly negatively correlated with TIL score, such that patients with a higher weighted Cam_121 gene expression score (standardized) had a poorer tumor infiltrating lymphocyte score (TIL count) and vice-versa. (A) Clark score from the AVAST-M primary tumours (n=137) B) Clark score from the LMC primary tumours (n=499) C) MIA score from the AVAST-M primary tumours (n=139). The global p-value was reported using ANOVA and the student's t-test was used for pairwise comparisons. The grey line represents the high vs low score cut-off at quantile 0.33.

2. Incorporating TIL count within the multivariate Cox regression analysis:

In light of the findings outlined above, the next question we sought to address was whether incorporating TIL count within the multivariate Cox regression survival analysis could alter the significance of the predictive signature. In particular, we wanted to assess whether the Cam_121 predictive signature adds any incremental prognostic information to a standard TIL count analysis, which as the reviewer has correctly pointed out, has probably long been recognised to be prognostically relevant in this context. As expected, Cam_121 was predictive of both overall and progression-free survival even after correction for TIL count in the AVAST-M primary melanoma data (MS Figure 1C). It was also predictive of melanoma specific survival after correction for TIL count in the Leeds Melanoma Cohort (MS Figure 2E).

Minor comments:

Top of page 10: Is „inputted“ really a verb?

We thank the reviewer for spotting this which we have corrected as below.

Previous: In order to identify the biological processes reflected by the signature, we inputted the genes into pre-ranked gene set enrichment analyses, ordered by their fold-change from the covariate-corrected differential expression analysis (see Methods).

Replaced with (now on page 12): In order to identify the biological processes reflected by the signature, we ran pre-ranked gene set enrichment analyses on the list of genes ordered by their shrinked log-fold change from the covariate corrected differential expression analysis (see Methods section 13.2)

Reviewer #2 (Remarks to the Author):

This study identified a gene signature that can predict long-term survival outcomes of patients with Melanoma from the transcriptome of a cohort where patients are dichotomized into two groups with either good or poor prognosis. Taking this gene signature and several clinical covariates, the authors build and select the best performing machine learning model with fine-tuned hyperparameters.

We thank the reviewer for their thorough assessment and helpful comments. The reviewer's inputs have encouraged us:

- To validate our machine learning (ML) results in an entirely separate dataset (MS Results section 3, Figure 3B).
- To clarify how ML was incorporated into our analyses - by including a machine learning workflow flowchart (Supplementary Figure S15 and Methods section 10).
- Present the results of the ML analyses using K-fold cross-validation, rather than leave-one-out cross-validation (LOOCV) (Supplementary Figure S7).
- To restructure the ML analyses within the Results section of the manuscript, clarifying that these analyses were used as an additional critical evaluation of the Cam_121 gene signatures' performance, rather than a signature optimisation strategy (Results section 2 & 3).

Before discussing the reviewer's points one-by-one, it seems useful to point out that

- our gene signature - generated, as the reviewer correctly pointed out, using differential expression analyses on the whole dataset - was critically evaluated in two ways:
 - (1) by means of Cox-regression survival models to assess the ability of the signature to predict progression-free and overall survival.
 - (2) by means of machine learning to assess the ability of the signature to predict metastases as an event.

Furthermore, the assessment of our signature on survival outcomes was performed both on the dataset used to define the signature as well as on four external validation datasets, including **The Cancer Genome Atlas** data (refer to our answer to point 2 of Reviewer 1), ie, on datasets independent of the one used to define the signature.

- machine learning classifiers were used to provide a critical evaluation of the signatures' performance, rather than choosing/optimising the signature definition.
- clinical covariates were not used to optimise the machine learning model, but were rather a comparator against which we critically evaluated the performance of our gene signature. In this way it is important to note that machine learning classifiers were trained separately; using either the clinical covariates or the gene signature or both (See Figure 3, panel: "Cam_121+Clinical Covariates"). Statistical testing was then undertaken to show the superior performance of the algorithm trained with gene signature relative to the algorithm trained with the clinical covariates alone, across all the machine learning classifiers (Supplementary Table S3).

This new classifier showed better performance when compared to existing melanoma signatures and machine learning models with clinical covariates only. They further validated this signature on other independent datasets. Overall, this study is important for the field of precision medicine and could lead to a clinically relevant signature for patient stratification that will reduce overtreatment and medical cost.

We thank the reviewer for their accurate summary of the study and its key findings. We entirely agree that such studies could be beneficial for patient stratification, which represents a particular area need in the current era with an ever rising incidence of early stage melanoma and effective adjuvant melanoma therapies (PMID's; 28822576, 28891423, 29477665). We greatly appreciate the helpful comments and methodological questions, which we have addressed point by point below.

However, there are several major concerns for the validity of the findings:

- 1. The 121-gene signatures were selected by using the whole dataset instead of leaving the validation sample out in each round of LOOCV. But feature selection should only be part of the training and not have validation set included. This is a major FLAW in training and testing supervised machine learning models. The selection of clinical variates seems to be dependent on the entire dataset as well, which again biased the selection results. LOOCV could be overly optimistic about the true accuracy. K-fold cross-validation should be used instead.**

We thank the reviewer for raising a number of critical points that we will now address point by point below. In summary, we have now validated our ML results in a completely independent dataset (AVAST-M lymph node dataset) which wasn't used for selecting any of the features (Figure R13; MS Results section 3, Figure 3B, Supplementary Figure S15, Methods section 10). We have also updated our cross validation (CV) method to use 10-Fold CV instead of LOOCV throughout the MS (MS Figure 3, Supplementary Figures S7 and S8, Methods section 10).

1. Feature selection was independent of machine learning

Firstly, it is important to point out as above that the 121-gene signature was selected using differential expression analyses and that machine learning was only used to evaluate its performance. In this way feature selection for the machine learning (the 121-gene signature) was undertaken prior to any machine learning training.

Similarly the clinical covariates were used as an entirely separate set of input features to train a separate machine learning model, which was used as a baseline model against which the performance of the Cam_121 gene signature was compared. The choice of these specific four clinical covariates was based on well-established clinical studies confirming the prognostic importance of these key factors used in melanoma staging (PMID: 29028110). In addition, and prior to inclusion of these specific covariates in the machine learning model, we undertook statistical testing to show that these clinical covariates were independently associated with metastases as an event (Methods section 8). We added a new ML flowchart which hopefully helps to clarify the ML workflow (Supplementary Figure S15).

2. Reproduction of ML results in a dataset not used for feature selection

We understand the reviewer's concern that despite performing nested cross-validation, the ML models may be giving superior performance as ultimately all the samples were used to select these features. In order to address this issue we have validated the results of the best performing classifier on each signature in an independent dataset. This was achieved using 177 regional lymph nodes from this trial, which represent an entirely separate set of samples to those with which the ML algorithm was trained. In doing this we observed the same trend in performance, with models trained with the signature alone performing better than models trained with the clinical covariates (Figure R3). We have further modified the ML workflow flowchart to clarify where this independent replication fits the wider ML analysis (Supplementary Figure S15).

Further external validation of the machine learning algorithm proved very challenging, as there are no suitable primary melanoma datasets which can be used to reliably replicate this approach. Specifically, all datasets with RNA sequencing in primary melanoma are not prospective or randomised and it would therefore not be possible to dichotomise patients from these studies into metastatic/non-metastatic cohorts in this way. Therefore reproducing the machine learning algorithm which is designed to predict metastases as an event was not feasible. We have now made reference to this in the discussion, highlighting the need for better quality data in this field (p.17).

However, we entirely agree that a separate validation dataset is nonetheless critical, and to this end we tested the signature's prognostic against survival outcomes using Cox regression models. These analyses validated in our findings in a separate internal dataset (Figure 2A-C), as well as in two external datasets (from the Leeds Melanoma Cohort and TCGA, Figure 2D-E and 2F-I respectively), with partial validation in two further datasets. These have now been specifically highlighted within the Results section 2 (p.7) as well as in Supplementary Figure S5 and highlighted in response to question 6 below.

Figure R13: ROC curves for validation of the best performing classifier for each signature obtained from Figure R14 below on the AVAST-M lymph node (LN) dataset. The p-values reported are obtained from DeLong's test for two

correlated ROC curves obtained by roc.test() function in R with alternative = "greater". Same as Figure 3B in manuscript.

3. K-fold cross validation

The reviewer makes a very important point regarding k-fold cross validation. In light of the reviewers' important comment, we have now trained our models using both 10-fold CV (repeated 1000 times) and LOOCV and directly compared the difference in model performance (Figure R14 below). We observed that both these methods yielded similar results during training, with k-fold CV giving marginally superior performance than LOOCV. Therefore, as recommended we have updated our ML results using 10-Fold CV (repeated 1000 times) (Figure 3A and Supplementary Figure S8). We further tested the performance of the final model from each signature on an independent LN dataset (Figure 3B-D) as described in the section above.

Figure R14: Plot showing the effect of the choice of cross validation on the classifier performance (assessed by area under the ROC curve values). Here, a set of 7 different classifiers is trained on each signature (shown as different panels) using 10-Fold cross validation (10-Fold CV) repeated 1000 times and leave-one-out cross validation (LOOCV) using the caret package in R.

The reviewers comments prompted us look deeper into the literature and a careful review of the properties of "Leave-one-out" (LOOCV) and "k-fold" cross-validations (KFCV) leads to conflicting conclusions depending on the context (for example, the type of models considered in the analysis [parametric versus non-parametric], the type of model selection [LASSO, AIC/BIC], the sample size of the full dataset ["small" versus "large"], the presence of heteroscedasticity, the level of overlap between training sets, etc.).

The popular/mainstream view is that LOOCV is less biased than KFCV when estimating the performance of a model (by means of the estimation of prediction errors) but has higher variability, leading to the usual bias-versus-variance trade-off (refer, for example, to Hastie et al

2017, p.242). Empirical evaluations of this trade-off, typically based on mean square error, led several authors to recommend using KFCV (like Hastie et al. 2017, Kohavi 1995, for example).

However, many authors also point out that, in some cases, LOOCV actually leads to the best/smallest mean-square error or that the advantages of KFCV over LOOCV are null/marginal. For example,

- Simulations and theoretical works of Burman (1989) and Lu (2007) for example showed that, in the case of the least-square regression, LOOCV actually both leads to the smallest bias and variance, when compared to other KFCVs.
- Simulations of Zhang and Yang (2015) showed that bias-versus-variance trade-off as well as root mean square error strongly depends on the method used to perform model selection [LASSO, AIC, aso].

References:

1. Kohavi, R., 1995. A study of cross-validation and bootstrap for accuracy estimation and model selection, In Proceedings of the 14th International Joint Conference on Artificial Intelligence, Vol. 2, Canada.
2. Hastie T, Tibshirani R, Friedman J: The elements of statistical learning, 2nd edition. New York: Springer; 2017.
3. Burman, P., 1989. A Comparative study of ordinary cross-validation, v-fold cross-validation and the repeated learning-testing methods. *Biometrika* 76, 503–514.
4. Lu, F. (2007) Prediction error estimation by cross validation (Ph.D. preliminary exam paper). School of Statistics, University of Minnesota.
5. Zhang, Y. and Yang, Y. (2015), Cross-validation for selecting a model selection procedure, *Journal of Econometrics*, v187, pp 95-112.

2. Does comparing the prediction performance on training dataset really support this idea? Since the model is trained on this dataset, there could be overfitting on the dataset but not performs equally well on any other external dataset. In addition, in the following validation on two external datasets, there seems to be no data showing the performance of classification but only survival analysis. The improvement over other methods on the two validation set is marginal. Is this due to hyperparam tuning? Are the other models tuned properly?

Overfitting within our dataset and lack of external validation of ML results

We thank the reviewer for this important point. As addressed in question 1 above, in addition to the training ML model, we have now performed an independent internal replication of our ML results on the AVAST-M lymph node dataset (Figure R13; MS Results section 3, Figure 3B, Supplementary Figure S15, Methods section 10). Unfortunately, as the external datasets only recorded outcome data on survival and were unable to dichotomise patients in metastatic and non-metastatic outcomes, we were unable to externally validate our ML analyses in these datasets. Instead we have now focussed more heavily on the external replication of the survival

analyses (see section “Reproduction of ML results in a dataset not used for feature selection” on page 21 above).

Hyperparameter tuning

Since the primary outcome available for these external studies were survival measures, we used the Cox Proportional-Hazards model to replicate these analyses. We would like to point out that there is no parameter/hyperparameter tuning involved when estimating Cox regression parameters. Indeed, an important property of the partial likelihood in the case of Cox regressions is that the baseline hazard is cancelled out of the equation, meaning that, under the model assumptions, the effect of covariates can be assessed without fitting further parameters (like the baseline hazard).

When comparing the Cox regression analysis across these different datasets, we note that different clinical covariates were taken into account within the multivariate model (fitted to each dataset separately). There was therefore further heterogeneity in terms of patient stage/length of follow-up/prospective vs retrospective study designs etc across the different validation datasets (see also answer to reviewer 1’s question 1 on page 4). However it is nonetheless important to point out that despite the aforementioned differences, the performance of our signature in terms of survival prediction remained consistent across these datasets.

3. The author didn’t show that the outperformance of gene signature over a model using only clinical variables is worth the extra cost for doing an assay for patients in practice though it’s statistically significant.

We agree with the reviewer that the cost-benefit balance of a new treatment/approach needs to be assessed carefully and that statistical significance may not correspond to a notable treatment gain. However, in our case, we believe that the potential benefits of our approach outweigh its cost because on top of its ability to predict survival and recurrence (which is critical for patients with Stage I/II), defining the Cam_121 signature has further benefits for the patients:

- a. this data would allow us to identify Stage II patients at high risk of metastases which is difficult to assess by looking at clinical covariates alone (MS Results section 5).
- b. we found our Cam_121 score to correlate with TIL counts (Figure R12; MS Results section 7: last paragraph; Figure 5) which are obtained by scoring TILs in H&E stained histopathological slides using different scoring schemes. This could lead to inter-pathologist as well as intra-pathologist biases in scoring where we found only 56% agreement between the 2 pathologists to assign a Clark score and only 40% agreement for MIA score. Thus, alongwith identifying patients with potentially worse outcomes, the Cam_121 gene expression score also shed light on the degree of tumor infiltrating lymphocytes in the primary tumor which can be used to guide tailored therapy to the patients.

Also, sequencing costs have strongly decreased during the last few years, especially as the Cam_121 signature can be obtained by testing a panel of 121 specific genes which is more cost effective than a full RNA seq.

In response to the reviewer's important point, we have added a sentence in the discussion attesting to the need for cost-benefit analyses in future studies (p.15): *'The challenge over the coming years will be to identify and validate a clinically-relevant measure of lymphocytic abundance of relevance to primary CM, that can be easily implemented in real-life clinical practice. These studies will also need to consider aspects of cost-effectiveness, which have not been explored in this analysis.'*

4. Criterion used to determine high/low signature expression: According to the method part, the cutoff for each dataset is the 0.33 quantile. It is a relative cutoff but not an absolute one. In practice, we can either assign a new patient its quantile in the training dataset or compare it with an absolute value. But during validation in these two external datasets, they are using 0.33 quantile cutoff in the new dataset, which could lead to a biased result.

We would like to thank the reviewer for pointing out potential risk of biased results due to the 0.33 quantile cutoff in both ours and the external datasets. To address this, we compared the 0.33 quantile cutoff values of the weighted Cam_121 signature across multiple datasets. We found that these cutoff values are actually very close to each other (cutoff close to -0.38 for samples derived from skin and cutoff close to -0.44 for samples derived from LN), see Figure R15 below. In response to these important comments made by the reviewer, we have added a statement in the methods attesting to the consistency of the 0.33 quantile expression cut-offs across the external validation datasets (Methods section 12, p.26). We would also like to point out that as part of the multivariate survival analysis, the weighted Cam_121 signature was used as a continuous variable in the Cox regression model. As such these results are independent of the choice of cutoff across these different datasets. We have now highlighted this both in the legends to these figures and the associated methods section (see Figures 1C, 2C, 2E, 2G and 2I (the forest plots) and Methods section 12).

Figure R15: Density distribution plot of weighted Cam_121 scores (standardized) derived from A) primary melanomas of the AVAST-M, LMC and TCGA-SKCM cohort. Note that for TCGA-SKCM Skin, samples are derived from regional cutaneous or subcutaneous tumours (see response to reviewer 1’s comment on page 13). B) Regional lymph node samples of the AVAST-M and TCGA-SKCM cohort. Each vertical line corresponds to the 0.33 quantile cutoff of the weighted gene expression signature in that cohort (A: AVAST-M Skin = -0.3826821, LMC Skin = -0.3896670, TCGA-SKCM Skin = -0.3420123 and B: AVAST-M LN = -0.4444574, TCGA-SKCM LN = -0.4378880).

5. The evaluation of clinical covariates in prediction power: the two groups of patients are classified based on the metastasis status while controlling for a ket set of clinical variables; thus it doesn’t reflect the real prediction power of clinical covariates between these two groups after controlling them. The concern is whether the clinical covariates really has a poor performance. The author may need to explain more on how they control the clinical variables.

We thank the reviewer for this important point. Firstly it is important to note that the four clinical covariates were selected entirely independently and based on longstanding clinical data confirming their prognostic relevance in this setting, as well as entirely separate statistical testing in this dataset confirming these covariates are also associated with poor outcomes in our study (specifically metastases and worse survival, see Table S1 and Methods section 8). In response to the reviewers’ earlier comments, we now further describe how these baseline clinical covariates were used as a comparator against which the performance of our gene signature was critically evaluated (MS Results section 3, Methods section 10). Specifically, we trained ML models using the baseline clinical covariates alone and compared their performance to the models trained with the gene signature alone (panel “Clinical covariates” in Figure 3A, Supplementary Figure S7 and S8 & Supplementary table S3). In doing this we show that the models trained with Cam_121 outperformed those trained with clinical covariates (Figure 3A, Supplementary Figure S7 and S8 and table S3). In response to these important comments, we have added a flowchart (Supplementary Figure S15) which we hope better clarifies this approach.

We did however control for these clinical covariates when assessing the effect of our signature in multivariate survival analyses. The results show that our signature has predictive power “on top” of the selected set of clinical covariates which had been shown to be significantly related to survival/relapse (MS Figures 1C, 2C, 2E, 2G, 2I).

6. The result from Lund primary melanoma cohort (the second external dataset) is not convincing: as the author mentioned that only 24 of 121 genes were included in this dataset, it is not able to demonstrate the prediction ability of the original Cam_121. What is the performance of these 24 genes on the other 2 datasets? Is LMC_150 completely included in all three datasets?

We thank the reviewer for highlighting this important observation and completely agree that given the paucity of genes within the Lund dataset, the survival results from this study does not fully reflect the predictive ability of Cam_121. It is important to consider that the low overlap in genes with the Lund dataset may be due to the different sequencing platforms, whereby the latter was acquired using an affymetrix array platform. This approach reported data on many fewer genes than our RNASeq platform, and was also biased towards genes that were much higher expressed (see Figure R2 in response to reviewer one’s first question above). In light of the reviewer’s valid comments, we have reduced the focus on this particular external validation cohort, and prefaced our discussion of the Lund melanoma cohort by highlighting that there was an insufficient number of overlapping genes for a complete validation (MS Results section 2, p.8). The performance of these 24 genes within the other datasets is reported in Figure R16 below. In regards to whether LMC_150 was completely included in all three datasets, we would like to highlight that only half (n=77) of the published LMC_150 genes were present in the Lund dataset (which was used as an external validation in that study, PMID: 31515461).

In view of this we sought to validate our signature in two further external datasets which had expression data on all 121 genes; The Cancer Genome Project (TCGA-SKCM, PMID: 26091043) and The Australia Melanoma Genome Project (AMGP, PMID: 28467829). The majority (219/472) of SKCM samples within the TCGA dataset are classified as ‘regional lymph nodes’ and subsetting to this sample type, we were able to validate our overall survival results in both the univariate (MS Figure 2F) and multivariate models (MS Figure 2G). There were only 87 primary melanoma samples in TCGA which was an insufficient sample size for this analysis (see also power calculation below), however when combining primary melanomas with 72 regional cutaneous relapses we were able to validate our findings again in both univariate and multivariate analyses (MS Figure 2H-I).

Unfortunately the AMGP RNASeq dataset (n=55) comprised a mixture of samples including; primary cutaneous melanomas, regional lymph nodes, regional cutaneous/subcutaneous relapses as well as distant metastases. Cox regression analyses from this dataset did confirm the same trend in performance (such that the high risk Cam_121 cohort demonstrated worse melanoma specific survival, Supplementary Figure S4), however this did-not reach statistical significance which is likely attributed to the mixed sample types as well as small sample size (as per power calculation outline in Figure R9 above). We have now included an explanation of this analysis within the results (p.8).

Notwithstanding the points above, we would still like to address the reviewer's question regarding the performance of these 24 genes in the LMC_150 and AVAST-M datasets. As shown in the Figure R16 below, although there was a trend in the expected direction such that a high signature score associated with a worse survival, these 24 genes did not survive multivariate analysis to predict progression free survival (PFS) within the AVAST-M primaries (top row, green) as well as melanoma specific survival (MSS) within the LMC primaries (bottom row, orange).

Figure R16: Forest plot indicating the hazard ratio estimates (point) and 95% confidence intervals (horizontal bars) related to the 24 overlapping genes of the Cam_121 signature with the Lund dataset when predicting PFS in AVAST-M primary melanoma dataset (top bar) and MSS in LMC dataset (bottom bar) by means of Cox proportional hazard models while controlling for different (sets of) clinical variables (y-axes). The Wald t-test p-value corresponding to the signature “Cam_24” parameter in each model is indicated on the top of each horizontal bar. ECOG; Eastern Cooperative Oncology Group Performance Status.

7. The comparison between Cam_121 and random 121 genes: the author uses a p-value to compare the performance of different features. However, since both OS and PFS are binary predictions, It is more common to compare the performance of different features by the AUROC.

We agree that the performance of the different ML models could be assessed in different ways. However we believe that our approach has advantages over the suggested alternative.

Our aim in this analysis is to assess whether our signature demonstrated an improvement over signatures based on random genes. To do this, we compare the p-value obtained by our signature to the distribution of p-values computed using randomly selected signatures in the same situation (i.e., same number of features, same sample size). P-values are a function of the test statistic which typically consists in the ratio of the parameter estimate and its measure of uncertainty (standard error). Therefore, our assessment relies on a function of the estimated log hazard ratios (corresponding to the signature parameter) standardised by their measure of uncertainty. As we considered the same situation in our comparisons (ie, same number of features, same sample size), the signature parameter standard errors were similar so that p-values were actually mostly a function of the (log) hazard ratios. This approach is commonly used for this purpose and is notably implemented in the Bioconductor package “SigCheck” (<https://www.bioconductor.org/packages/release/bioc/html/SigCheck.html>), designed to assess the uniqueness and utility of gene signatures.

The suggested alternative, relying on AUROC comparisons, is implementable, but, to our perception, has drawbacks compared to the *SigCheck-like* approach described above. Indeed, the output of the Cox model enables estimation of the probability of survival for all patients given their signature (continuous or dichotomous) at a given time point. To define the AUROC, these probabilities need to be “rounded” at a given time point to obtain binarised outcomes (dead/alive for the outcome “overall survival” and recurrence/no recurrence for the outcome “progression free survival”) which are to be compared to the observed events at the same time. Due to differences in follow-up times, several time points are possible. Thus,, as the AUROC approach relies on a binarized prediction (loss of information) at a given time point (several options), we believe that our P-value based approach, also commonly used, may be better suited in this context.

8. Paragraph 4 of section 6.2 is very confusing. It reads like they trained the model using nested-LOOCV for each classifier using each signature. However, according to the context, the models were trained using all the selected signatures and the clinical covariates.

We apologise for the confusion and have rewritten the MS Results section 3 and Methods section 10, now reporting 10-Fold CV rather than nested-LOOCV. We have also highlighted that the training was undertaken on skin primaries with validation on a separate LN dataset. The reviewer is correct in that for each signature, 7 different classifiers were trained using cross-validation and the best performing classifier in terms of AUROC was selected as the final model for testing. We hope that the new ML flowchart helps to clarify this further (Supplementary Figure S15).

Minor editorial issues like typos or format issues:

-Page 19-21, wrong item number 6.x after 8.

-Page 44 : The p-values are defined as the fraction of scores of random signatures which are greater THAN the one observed when using the (real) Cam_121 gene signature.

We thank the reviewer for highlighting these, which we have corrected within the manuscript.

Reviewers' Comments:

Reviewer #1:

Remarks to the Author:

The authors have diligently addressed all the concerns of this reviewer and gone beyond what was expected. They have significantly improved the paper and its significance.

Reviewer #2:

Remarks to the Author:

The authors have addressed most of my concerns except the feature selection. This is most critical as it can cause overfitting. In the author response, they said "121-gene signature was selected using differential expression analyses and that machine learning was only used to evaluate its performance." This is precisely what I am worried about. The feature was selected on the training and testing samples. Please see this post

<https://stats.stackexchange.com/questions/64825/should-feature-selection-be-performed-only-on-training-data-or-all-data>

Reviewer #2 (Remarks to the Author):

The authors have addressed most of my concerns except the feature selection. This is most critical as it can cause overfitting. In the author response, they said "121-gene signature was selected using differential expression analyses and that machine learning was only used to evaluate its performance." This is precisely what I am worried about. The feature was selected on the training and testing samples. Please see this post <https://stats.stackexchange.com/questions/64825/should-feature-selection-be-performed-only-on-training-data-or-all-data> [stats.stackexchange.com]

We thank the reviewer for their extremely helpful review and valuable comments. We entirely agree that this is a critical point and have carefully considered this as outlined below.

We would like to start by pointing out that our Cam121 signature has been extensively validated across several external validation cohorts. External validation is more stringent than internal validation as it tests the final model in a setting fully independent of the one used to develop the model (Harrell, 2015, Section 5.3.1). In addition, prognostic signatures are generally evaluated on their ability to predict patients' survival outcomes, which is why we initially focussed our evaluation on the Cox regression survival analyses of external cohorts. The results we obtained in the two external cohorts that had sufficient sample size to detect the effect sizes observed in the initial cohort with a high probability (Supplementary Figure S4), the *Leeds* (Figures 2D&E) and the *TCGA* cohorts (Figures 2E-I), are similar to the ones observed in the original AVAST-M Skin cohort (n=194) (Figures 1A-C). We believe that this alone provides a proof that our signature has the prognostic power we claim.

Regarding the validation using machine learning approaches, after careful consideration, we agree with the reviewer that the internal validation initially presented in Figure 3A was not optimal. Firstly, the LMC_150 and DecisionDx-Melanoma models, developed on different cohorts, were initially compared to signatures optimised on the same data used to generate Figure 3A. Secondly, as the initial internal validation in Figure 3B did-not consider the uncertainty related to the model selection in Figure 3A, it may be overoptimistic (Krstajic et al, 2015, page 2).

Consequently, to specifically address the potential overfitting issue, we have taken the following actions:

- First, we removed the LMC_150 and DecisionDx-Melanoma models from **Figure 3A (as well as from Supplementary Table S3A-B, Supplementary Figures S7 and S8)** and therefore we no longer compare the LMC_150 and DecisionDx-Melanoma models to models optimised on the AVAST-M Skin dataset.
- Second, we now only use **Figure 3A** to select the best classifier for the 'Cam_121 + Clinical Covariates', 'Cam_121' and 'Clinical Covariates' models that we then consider further, on another entirely independent cohort, the AVAST-M Lymph node samples (n=143), in **Figure 3B**, to compare their respective AUROC statistics. Thus, the aim of the cross-validation is not to perform a model validation in the AVAST-M Skin cohort

but rather to select a classifier that is then tested on an independent dataset. Note that such a "*model validation is not necessary unless the analyst wishes to use it to quantify the degree of overfitting*" (Harrell, 2015, Section 4.12.3) which is not our aim here. In agreement with the reviewer's comments, on page 9 of the manuscript we have added a comment that the AUROC estimates are likely to be over-optimistic, however we do-not believe this is an issue for selecting the classifier of interest.

- Third, in a **new analysis** presented in **Figure 3E** (and shown in Figure 1 below), we now compare the 5 models of interest, ie., (i) 'Cam_121 + Clinical Covariates', (ii) 'Cam_121', (iii) 'Clinical Covariates', (iv) LMC_150 and (v) DecisionDx-Melanoma on the independent cohort, the AVAST-M Lymph node samples. Thus, the assessment of each model of interest is now performed on a data set that is independent of the one used to define them. This approach is summarised on page 10 of the manuscript. The AVAST-M Skin and Lymph node datasets are a separate set of samples derived from the same phase III trial. As the signature was defined on the Skin cohort and is now tested on the Lymph node, the analysis of **Figure 3E** may be considered as an internal validation considering a *data splitting* by disease site (refer to Harrell, 2015, Section 5.3.3.).
- Furthermore, we applied two different resampling methods to assess the properties of the models of interest (**Figure 3E**):
 - a nested repeated 10-fold cross validation (indicated in orange). Krstajic et al (2014) described this CV, in its stratified version or not, as the most suitable to perform "cross-validators assessment" (as described by Stone, (1974))
 - a non-parametric bootstrap (indicated in green). This resampling method is often described as "preferred" validation method (Moons et al, 2012, p. 687; Harrell, 2015, Section 5.3.5)

Both methods lead to the same conclusions: models based on the Cam_121 signature perform better than the other models (Figure 3E and Supplementary Table 3C - the latter highlighting the comparative statistical testing).

We have updated the methodological details of this new analysis within the Methods section 10. As a side note, we also spotted a typographical error in the group labelling of Figure 1C and have corrected it accordingly.

Figure 1: Dotplot showing the performance of different classification models in predicting metastases in terms of AUROC (mean \pm standard deviation) when trained on the AVAST-M lymph node dataset (n=143). Within each panel, 14 different machine-learning classifiers were trained to predict metastases: 7 using 10-Fold CV (repeats = 1000) and 7 using bootstrap resampling method (repeats = 1000). The 2 horizontal lines indicated within each panel denote the median AUROC of these 7 classifiers respectively. Statistical comparison using student's t-test are indicated (bootstrap in green and 10-Fold CV in orange). Decision-Dx Melanoma: Decision-Dx MelanomaTM, LMC_150: Leeds Melanoma Cohort 150 gene signature.

References:

- Harrell, F. (2015), Regression Modeling Strategies, Springer.
- Krstajic, D., Buturovic, L.J., Leahy, D. and Simon T. (2014), Cross-validation pitfalls when selecting and assessing regression and classification models, Journal of Cheminformatics, 6:10.
- G M Moons, K.G.M., Kengne, A.P., Woodward, M., Royston, P., Vergouwe, Y., Altman, D.G. and Grobbee, D.E. (2012), Risk prediction models: I. Development, internal validation, and assessing the incremental value of a new (bio)marker, Heart, V. 98, p. 683-690
- Stone, M. (1974), Cross-Validatory Choice and Assessment of Statistical Predictions, Journal of the Royal Statistical Society Series B, V. 36, N. 2, p. 111-147.

Reviewers' Comments:

Reviewer #2:

Remarks to the Author:

All concerns are addressed